



# Zooplankton mortality effects on the plankton community of the Northern Humboldt Current System: Sensitivity of a regional biogeochemical model

Mariana Hill Cruz[1], Iris Kriest[1], Yonss Saranga José[1], Rainer Kiko[2], Helena Hauss[1,3], and Andreas Oschlies[1,3]

[1]GEOMAR Helmholtz Centre for Ocean Research Kiel, Düsternbrooker Weg 20, 24105 Kiel, Germany
[2]Sorbonne Université, Laboratoire d'Océanographie de Villefranche-sur-mer, France
[3]Christian-Albrechts-University Kiel, Germany

**Correspondence:** Mariana Hill Cruz (mhill-cruz@geomar.de)

**Abstract.**

Small pelagic fish off the coast of Peru in the Eastern Tropical South Pacific (ETSP) support around 10% of the global fish catches. Their stocks fluctuate interannually due to environmental variability which can be exacerbated by fishing pressure. Because these fish are planktivorous, any change in fish abundance may directly affect the plankton and the biogeochemical system.

To investigate the potential effects of variability in small pelagic fish populations on lower trophic levels, we used a coupled physical-biogeochemical model to build scenarios for the ETSP and compare these against an already published reference simulation. The scenarios mimic changes in fish predation by either increasing or decreasing mortality of the model's large and small zooplankton compartments.

The results revealed that large zooplankton was the main driver of the response of the community. Its concentration increased under low mortality conditions and its prey, small zooplankton and large phytoplankton, decreased. The response was opposite, but weaker, in the high mortality scenarios. This asymmetric behaviour can be explained by the different ecological roles of large, omnivorous zooplankton, and small zooplankton, which in the model is strictly herbivorous. The response of small zooplankton depended on the antagonistic effects of mortality changes as well as the grazing pressure by large zooplankton. The results of this study provide a first insight on how the plankton ecosystem might respond if variations in fish populations were modelled explicitly.

## 1 Introduction

Eastern Boundary Upwelling Systems (EBUS) are among the most productive regions in the ocean. Despite their small size, they support a large fraction of the world's fisheries (Chavez and Messié, 2009). The Northern Humboldt Current System (NHCS) in the Eastern Tropical South Pacific (ETSP) Ocean is the most productive EBUS, producing 10% of the global fish catches (Chavez et al., 2008), and supporting the fishery of the Peruvian anchovy *Engraulis ringens*, which is the biggest





single-species fishery on the planet (Chavez et al., 2003). The ETSP is also characterised by substantial inter-annual variability (i.e., El Niño Southern Oscillation, Holbrook et al., 2012), and an intense midwater Oxygen Minimum Zone (OMZ), resulting in high denitrification rates (Farías et al., 2009).

As in other EBUS, small pelagic fish are highly abundant (Cury et al., 2000) in the NHCS, building up large populations that are severely affected by climate fluctuations. For example, anchovy biomass in the NHCS fluctuated between 10 and 16 million tonnes in the 1960's (Alheit and Niquen, 2004). Its area of distribution spans from northern Peru to northern Chile and the Telcahuano region off Central Chile (Figure 1 in Alheit and Niquen, 2004). During the El-Niño event of 1972, it dropped to 6 million tonnes (Alheit and Niquen, 2004), presumably due to sensitivity to environmental variability exacerbated by fishing
pressure (Beddington and May, 1977; Hsieh et al., 2006). From 1992 to 2008, the population of anchovy off the Peruvian coast fluctuated between 3 and 12 million tonnes (Figure 13 in Oliveros-Ramos et al., 2017).

The Peruvian anchovy is a planktivorous fish. Its first-feeding larvae consume mainly phytoplankton. When they reach a length of 4 mm their diet gradually changes to zooplankton, especially nauplii of copepods (Muck et al., 1989). Adult anchovies still consume phytoplankton but their main source of energy are euphausiids and copepods (Espinoza and Bertrand,
2008). Pacific sardine *Sardinops sagax*, the second prominent small pelagic fish species in the NHCS, feeds on phytoplankton and small zooplankton (Ayón et al., 2008a). These two dominant species can therefore be expected to impose a direct top-down control on plankton; at the same time, they may be bottom-up affected by changes in plankton abundance caused by variations in physical forcing.

Pauly et al. (1989) estimated that the total population of anchovy off Peru consumes 12.1 times its own biomass in one
year. Assuming an area of $6 \times 10^{10}$ m$^2$ (Ryther, 1969) and a conversion factor of zooplankton wet weight to nitrogen of 1000 mg ww (mmol N)$^{-1}$ (Travers-Trolet et al., 2014a), a fluctuation in anchovy population of 9 Mt would result in a change in zooplankton mortality of 5 mmol N m$^{-2}$ d$^{-1}$ from anchovy predation alone, in a top-down driven ecosystem. The assumption that anchovy can exacerbate a top-down control on zooplankton is supported by a decline of zooplankton concentration in dense aggregations of anchovies (Ayón et al., 2008a, b). On the other hand, co-occurring long-term fluctuations of zooplankton
and anchovies at population scale indicate also a relevant bottom-up control in the NHCS (Alheit and Niquen, 2004; Ayón et al., 2008b).

Numerical models are valuable tools to examine the potential tight coupling across a large range of trophic levels and the mutual interactions among the different components, including top-down and bottom-up effects. Rose et al. (2010) pointed out the increasing need for so-called end-to-end models of the marine food webs, which couple models including physical and
biogeochemical processes with models for higher trophic levels. When coupled, organisms (plankton) of the former typically provide the food for the latter (e.g., Travers-Trolet et al., 2014b).

In stand-alone biogeochemical models that do not include higher trophic levels, zooplankton mortality is a closure term, used to return the additional biomass to detritus. It represents all processes that reduce the concentration of zooplankton and are not explicitly included in the model (for instance, predation by gelatinous organisms, predation by higher trophic levels and
non-consumptive mortality). For example, Getzlaff and Oschlies (2017) used it to mimic predation and immediate egestion



or mortality by higher trophic levels. Zooplankton mortality may also form the link to fish models, when these are explicitly considered in the context of biogeochemical models (e.g., Travers-Trolet et al., 2014b).

However, there is no consensus on the form of the mortality term, linear and quadratic being two common forms (e.g. Evans and Parslow, 1985; Fasham et al., 1990; Koné et al., 2005; Kishi et al., 2007; Aumont et al., 2015). A common argument for preferring quadratic to linear mortality is the reduction of unforced short-term oscillations (Steele and Henderson, 1992), although Edwards and Yool (2000) argue that quadratic mortality does not always remove such oscillations. A quadratic mortality term may also be interpreted as an increase in diseases because of high population densities, cannibalism, or increased predation due to high densities of prey. Because it is very difficult to determine zooplankton mortality in the field, there is also no agreement on the exact value of mortality (either linear or quadratic), and this term, in practice, is often adjusted to tune the model. However, not using mortality rates based on observations may limit the capability of the model for an accurate representation of the zooplankton compartment (Daewel et al., 2013), and to draw predictions about the state and dynamics of the marine ecosystem (Anderson et al., 2010). Hirst and Kiørboe (2002) predicted a global mortality of copepods of 0.062 d$^{-1}$ at 5°C and 0.19 d$^{-1}$ at 25°C in the field. Two thirds of such mortality are due to predation. In models, the values of quadratic mortality rate (hereafter called $\mu_Z$) in the literature vary over a large range, from 0.025 (mmol N m$^{-3}$)$^{-1}$ d$^{-1}$ (Fennel et al., 2006) up to 0.25 (mmol N m$^{-3}$)$^{-1}$ d$^{-1}$ (Lima and Doney, 2004).

For the NHCS, the high variability in forcing, biogeochemistry, plankton, and the high abundance of planktivorous fish, together with its economic importance indicates the need for end-to-end models that include details of all components. However, developing such a model is challenging and studies on this region have so far either focused on fish (Oliveros-Ramos et al., 2017) or physics and biogeochemistry (Jose et al., 2017). Given the large importance of zooplankton mortality as a link between these two model systems (Mitra et al., 2014), and the uncertainty associated with it, in this study we focus on the effects of this parameter on the biogeochemical system of the ETSP as a first step towards a fully coupled system.

In order to model the highly dynamic nature of both physical and biogeochemical processes in the ETSP, we employed a biogeochemical model specifically designed for EBUS, coupled to a finely resolved regional circulation model. The coupled model has already been validated against oxygen, nitrate and chlorophyll (Jose et al., 2017); this configuration serves as a starting point and reference for the sensitivity experiments. We first extend the model validation by Jose et al. (2017) and assess the reliability of the zooplankton by comparing the simulated mesozooplankton concentrations against observations obtained by net hauls. We then present two sensitivity experiments in which we varied the mortality rate of quadratic zooplankton mortality by ±50 %. Model sensitivity is examined with regard to concentrations of model components and inter-compartmental fluxes. Besides the overall response of the prognostic variables to an increase or decrease in zooplankton mortality, we describe how changing zooplankton mortality affects the trophic structure of plankton, with focus on the highly productive coastal domain. Finally we discuss the implications of our study for the plankton community and for modelling higher trophic levels.



## 2 Methods

### 2.1 ROMS-BioEBUS model set up

The Regional Oceanic Modeling System (ROMS; Shchepetkin and McWilliams, 2003, 2005) is a high resolution, free-surface,
90 terrain-following coordinate ocean model that solves the primitive equations considering the Boussinesq and hydrostatic
assumptions. A Biogeochemical model for Eastern Boundary Upwelling Systems (BioEBUS), which was derived from a
$N_2P_2Z_2D_2$ model from Koné et al. (2005), is coupled online to the physical part (Gutknecht et al., 2013a, b).

In this study the coupled ROMS-BioEBUS model has the same configuration as in Jose et al. (2017). It contains a small,
high resolution domain forced by a larger coarse resolution domain, using the AGRIF 2-WAY nesting procedure. The small
95 inner grid has a horizontal resolution of 1/12° spanning from about 69° W to 102° W and from 5° N to 31° S (Figure 1). The
large outer grid spans from 69° W to 120° W and from 18° N to 40° S with a resolution of 1/4°. The biological processes
occur in three time steps of 900 seconds for each physical time step of 2700 seconds in both domains. The two domains have
32 sigma layers with a vertical resolution of less than 5 m at the surface and decreasing to around 500 m at a maximum depth
of 4500 m.

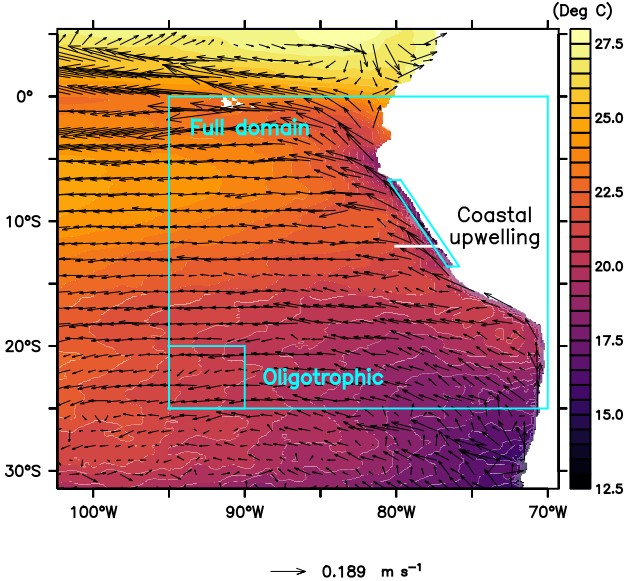

**Figure 1.** Annually averaged sea surface temperature (° C) and horizontal advection vectors (m s$^{-1}$), and location of the analysed regions
(see labels), and of a vertical section for analysing plankton spatial succession (white line at 12° S).

The coarse-resolution domain temperature, salinity and horizontal velocity are forced at the lateral boundaries with monthly
climatological (1990-2010) SODA reanalysis (Carton and Giese, 2008). Both domains are forced at the surface with 1/4°
resolution wind velocity fields from QuikSCAT (Liu et al., 1998) and monthly heat and freshwater fluxes from COADS (Worley
et al., 2005). At the lateral boundaries of the coarse resolution domain, the biogeochemical model is forced with monthly





nitrate and oxygen values from CARS (Ridgway et al., 2002) and surface chlorophyll from SeaWiFs (O'Reilly et al., 1998).
Phytoplankton and zooplankton boundary conditions are derived from a vertical extrapolation of the chlorophyll data. Detailed information about the boundary and initial conditions, and validation of the model is available in Jose et al. (2017).

## 2.2 Biogeochemical model description

The evolution of a biological tracer in time is represented by Eq. (1). On the right-hand side of the equation, the first term represents the advection with the velocity vector $u$. The eddy and molecular diffusion is represented by the second and third
terms, where $K_h$ is the horizontal diffusion coefficient and $K_z$ the vertical diffusion coefficient. The last term is a source-minus-sink ($SMS$) term due to biological processes. The full set of equations and detailed explanation about each process are available in Gutknecht et al. (2013a).

$$\frac{\partial C_i}{\partial t} = -\nabla \cdot (uC_i) + K_h \nabla^2 C_i + \frac{\partial}{\partial z}(K_z \frac{\partial C_i}{\partial z}) + SMS(C_i) \tag{1}$$

The BioEBUS model is adapted for the biogeochemical processes specific to the low-oxygen conditions of EBUS, with
some processes being oxygen dependent (see Gutknecht et al., 2013a). It has two compartments of phytoplankton (nanophytoplankton $P_S$, e.g. flagellates, and microphytoplankton $P_L$, e.g. diatoms), two compartments of zooplankton (microzooplankton $Z_S$, e.g. ciliates, and mesozooplankton $Z_L$, e.g. copepods), which will be described in detail below, and two compartments of detritus (small $D_S$ and large $D_L$). The two size classes of detritus are produced from phytoplankton and zooplankton mortality, and by release of unassimilated grazing material. The model also contains three compartments of dissovled inorganic nitrogen
($NH_4^+$, $NO_2^-$ and $NO_3^-$), dissolved organic nitrogen (DON), oxygen ($O_2$) and nitrous oxide ($N_2O$).

Remineralization processes are divided into ammonification, nitrification and denitrification, and are based on the formulations by Yakushev et al. (2007). $N_2O$ is a diagnostic variable for model output and its production does not affect the concentration of the other variables. It is based on the parameterization of Suntharalingam et al. (2000, 2012), which relates the production of $N_2O$ to the consumption of $O_2$ from decomposition of organic matter in oxic and suboxic conditions. $O_2$ con-
centrations depend on primary production, zooplankton respiration, nitrification and remineralization. This model includes gas exchange of $O_2$ and $N_2O$ with the atmosphere.

### 2.2.1 Phytoplankton

The $SMS$ terms in the small and large phytoplankton compartments are determined by Eq. (2) and (3), respectively:

$$SMS(P_S) = (1 - \varepsilon_{P_S}) \cdot J_{P_S}(PAR, T, N) \cdot [P_S] - G_{Z_S}^{P_S} \cdot [Z_S] - G_{Z_L}^{P_S} \cdot [Z_L] - \mu_{P_S} \cdot [P_S] \tag{2}$$

$$SMS(P_L) = (1 - \epsilon_{P_L}) \cdot J_{P_L}(PAR, T, N) \cdot [P_L] - G_{Z_S}^{P_L} \cdot [Z_S] - G_{Z_L}^{P_L} \cdot [Z_L] - \mu_{P_L} \cdot [P_L] - w_{P_L} \cdot \frac{d}{dz}[P_L]. \tag{3}$$





Where $J_{\mathrm{P}_i}(\mathrm{PAR}, T, \mathrm{N})$ is the growth rate, limited by light, temperature, and nutrients, and $\epsilon_{\mathrm{P}_i}$ is the exudation fraction of primary production;

$G_{\mathrm{Z}_j}^{\mathrm{X}_i}$ are feeding rates by zooplankton (see Section 2.2.2), $\mu_{\mathrm{P}} \cdot [\mathrm{P}]$ is the mortality term, representing all not explicitly modelled phytoplankton losses. Large phytoplankton is characterised by a steeper initial slope of the P-I curve and by larger

half-saturation constants for nutrient uptake (see Table A1). Therefore, it grows better than small phytoplankton under low light conditions, but its nutrient uptake increases more slowly as nutrient concentrations increase. Finally, large phytoplankton is subjected to an additional loss from sedimentation, via a constant sinking speed of $w_{\mathrm{P}_{\mathrm{L}}}$ (Gutknecht et al., 2013a).

### 2.2.2 Zooplankton

Zooplankton increases its biomass through grazing on phytoplankton, and in the case of large zooplankton also on small

zooplankton. Metabolism, mortality and, in the case of small zooplankton, predation by large zooplankton are sink terms. Predation by fish and other higher trophic levels is implicit in the quadratic mortality term. The biomass lost by metabolism and mortality is assumed to become detritus which may sink to the sediments or become remineralized, and a small fraction of zooplankton losses becomes part of the DON pool which is also subjected to remineralization.

The $SMS$ terms of the small and large zooplankton compartment are determined by Eq. (4) and (5), respectively:

$$SMS(\mathrm{Z}_{\mathrm{S}}) = f1_{\mathrm{Z}_{\mathrm{S}}} \cdot (G_{\mathrm{Z}_{\mathrm{S}}}^{\mathrm{P}_{\mathrm{S}}} + G_{\mathrm{Z}_{\mathrm{S}}}^{\mathrm{P}_{\mathrm{L}}}) \cdot [\mathrm{Z}_{\mathrm{S}}] - G_{\mathrm{Z}_{\mathrm{L}}}^{\mathrm{Z}_{\mathrm{S}}} \cdot [\mathrm{Z}_{\mathrm{L}}] - \gamma_{\mathrm{Z}_{\mathrm{S}}} \cdot [\mathrm{Z}_{\mathrm{S}}] - \mu_{\mathrm{Z}_{\mathrm{S}}} \cdot [\mathrm{Z}_{\mathrm{S}}]^2 \tag{4}$$

$$SMS(\mathrm{Z}_{\mathrm{L}}) = f1_{\mathrm{Z}_{\mathrm{L}}} \cdot (G_{\mathrm{Z}_{\mathrm{L}}}^{\mathrm{P}_{\mathrm{S}}} + G_{\mathrm{Z}_{\mathrm{L}}}^{\mathrm{P}_{\mathrm{L}}} + G_{\mathrm{Z}_{\mathrm{L}}}^{\mathrm{Z}_{\mathrm{S}}}) \cdot [\mathrm{Z}_{\mathrm{L}}] - \gamma_{\mathrm{Z}_{\mathrm{L}}} \cdot [\mathrm{Z}_{\mathrm{L}}] - \mu_{\mathrm{Z}_{\mathrm{L}}} \cdot [\mathrm{Z}_{\mathrm{L}}]^2. \tag{5}$$

Here $f1_{\mathrm{Z}_{\mathrm{S}}}$ and $f1_{\mathrm{Z}_{\mathrm{L}}}$ are assimilation coefficients (see also Table A1). $G_{\mathrm{Z}_j}^{\mathrm{X}_i}$ are feeding rates of predator $\mathrm{Z}_j$ (either large or small zooplankton) on prey $\mathrm{X}_i$ (small and large phytoplankton, and small zooplankton) calculated with the formulation by Tian et al. (2000, 2001). There is a linear loss rate accounting for basic metabolism ($\gamma_{\mathrm{Z}_i}$), and a quadratic loss rate also referred

as mortality. The mortality parameters $\mu_{\mathrm{Z}_{\mathrm{S}}}$ and $\mu_{\mathrm{Z}_{\mathrm{L}}}$ of the reference simulation are 0.025 and 0.05 (mmol N m$^{-3}$)$^{-1}$ d$^{-1}$ for small and large zooplankton, respectively, as in Jose et al. (2017) and Gutknecht et al. (2013a).

The feeding rate follows the formulation from Tian et al. (2000, 2001):

$$G_{\mathrm{Z}_j}^{\mathrm{X}_i} = g_{\mathrm{max}_{\mathrm{Z}_j}} \cdot \frac{e_{\mathrm{Z}_j \mathrm{X}_i} \cdot [\mathrm{X}_i]}{k_{\mathrm{Z}_j} + F_t} \tag{6}$$


where $g_{\mathrm{max}_{\mathrm{Z}_j}}$ is the maximum grazing rate of predator $\mathrm{Z}_j$, $e_{\mathrm{Z}_j \mathrm{X}_i}$ is the preference of predator $\mathrm{Z}_j$ for prey $\mathrm{X}_i$. $k_{\mathrm{Z}_j}$ is the half-saturation constant and $F_t$ is the total availability of food for predator $\mathrm{Z}_j$. Large zooplankton responds more slowly to changes in food due to the high $k_{\mathrm{Z}_{\mathrm{L}}}$. In the case of large zooplankton $F_t = e_{\mathrm{Z}_{\mathrm{L}} \mathrm{P}_{\mathrm{S}}} \cdot [\mathrm{P}_{mathrmS}] + e_{\mathrm{Z}_{\mathrm{L}} \mathrm{P}_{\mathrm{L}}} \cdot [\mathrm{P}_{mathrmL}] +$





$e_{Z_L Z_S} \cdot [Z\ _{mathrmS}]$ and in the case of small zooplankton $F_t = e_{Z_S P_S} \cdot [P\ _{mathrmS}] + e_{Z_S P_L} \cdot [P\ _{mathrmL}]$ (Gutknecht et al.,

2013a).

### 2.3    Zooplankton validation

As noted above, this model was already validated against oxygen, nitrate and chlorophyll (Jose et al., 2017). As a complement, we here compare the large zooplankton compartment of the model with observational data collected on cruise M093-1.The samples obtained during this cruise include day- and nighttime hauls with a Hydrobios multinet (nine nets, 333 $\mu$m mesh)

between February 10 to March 3, 2013 on a transect off the Peruvian coast ($\approx 12°$S; see Fig. 2f), capturing the vertical and horizontal gradient in zooplankton concentration. Samples were size-fractionated by sieving and processed according to the ZooScan method (Gorsky et al. (2010). Observations included crustaceans, chaetognaths and annelids greater than 500 $\mu$m. For model comparison we converted the observation from nighttime hauls to dry biomass according to Lehette and Hernández-León (2009), and further to nitrogen units as suggested by Kiørboe (2013). A detailed description of the zooplankton processing

is provided by Kiko and Hauss (2019). Only night observations were compared since our model does not include diel vertical migration.

Both in the model and observations, concentration of large zooplankton is greatest in the surface and decreases with depth (Figure 2). At the surface, modelled concentrations are 1 order of magnitude larger than observations at almost all stations (Figure 2). Only at station d), observations reach 1 mmol N m$^{-3}$, while the model exhibits maximum values close to 4 mmol N m$^{-3}$.

At most stations, the distribution of modelled concentrations is similar to observed concentrations in the surface layer (upper 100 m), although model estimates are consistently higher. Below 100 m, however, model estimates are consistently lower than the observations, which is in particular evident at the deep offshore stations (Appendix B, Figure B1 a) and b)). Zooplankton in our model does not consume detritus or bacteria; small zooplankton feeds on phytoplankton, and large zooplankton feeds on small zooplankton and on phytoplankton. Therefore, in contrast to observations, its presence is not expected in deep water.

In summary, the model matches the observed spatial pattern of zooplankton distribution, but tends to overestimate zooplankton concentration in the surface layer and to underestimate it in the mesopelagic. Possible reasons for this mismatch will be discussed in Sect. 4.1.

### 2.4    Experimental design

To mimic changes in grazing pressure on zooplankton due to small pelagic fish biomass fluctuations, we followed the approach

by Getzlaff and Oschlies (2017), and varied the mortality rate of each zooplankton compartment by $\pm 50$ % in comparison to the reference scenario described by Jose et al. (2017). Thereby, an increase in mortality assumes a large consumption of zooplankton by fish, while a decrease in mortality assumes less fish. Because the model does not include an explicit compartment for fish, it is assumed that all zooplankton biomass consumed by fish becomes part of the detritus pool via immediate fish mortality and defecation. In reality, a fraction of the biomass is extracted from the system by the fishing industry, predation of

sea birds that defecate over land, and migrations.

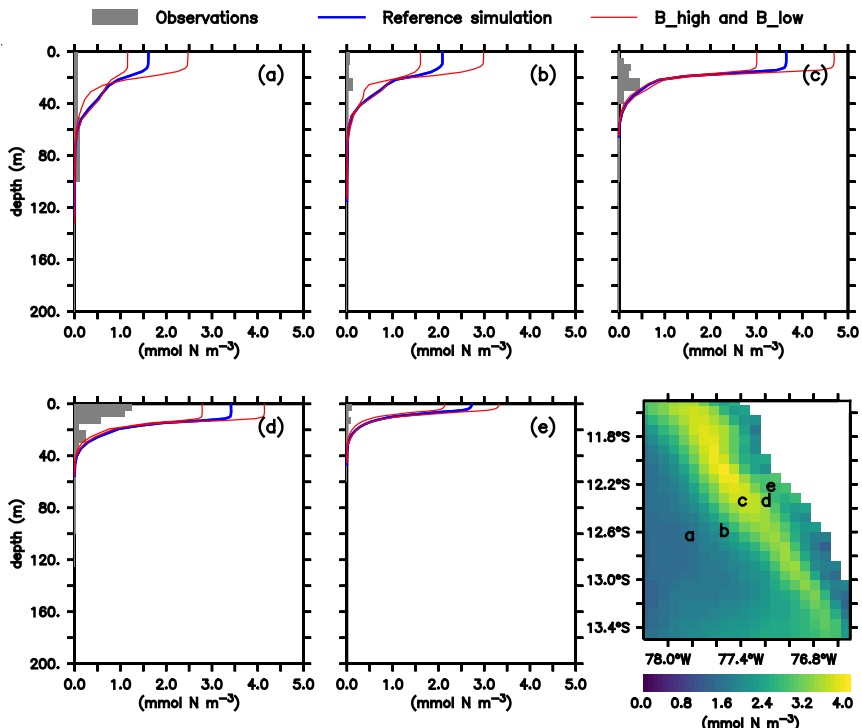

**Figure 2.** a) to e) Mesozooplankton concentrations (mmol N m$^{-3}$). Lines indicate modelled large zooplankton concentrations in the reference scenario and experiments B_high and B_low. Shaded area shows observed nighttime mesozooplankton biomass concentrations over the sampled depth intervals (m). Observations are lower than 0.1 mmol N m$^{-3}$ below 200 m, thus they have not been included. For a plot including deep water observations, please see Appendix B, Fig., B1, and Fig. 4 in Kiko and Hauss (2019). Bottom right: Modelled large zooplankton biomass concentration at the surface in the reference scenario (mmol N m$^{-3}$) and locations where observations were collected.

Our model has two zooplankton compartments. In order to explore the different roles of large zooplankton as top predator and small zooplankton as grazer and prey, we performed four experiments, in which we varied the respective mortality rate of large and small zooplankton (0.05 and 0.025 [mmol N m$^{-3}$]$^{-1}$ d$^{-1}$, respectively) by ±50 %:

- A_high with $1.5 \times \mu_{Z_L}$

- A_low with $0.5 \times \mu_{Z_L}$

- B_high with $1.5 \times \mu_{Z_L}$ and $1.5 \times \mu_{Z_S}$

- B_low with $0.5 \times \mu_{Z_L}$ and $0.5 \times \mu_{Z_S}$

where $\mu_{Z_i}$ are the mortality rates of large and small zooplankton. The average nitrogen flux to detritus due to large zooplankton mortality over the upper 100 m depth near the coast of Peru (coastal upwelling region, see Section 2.5 and Figure





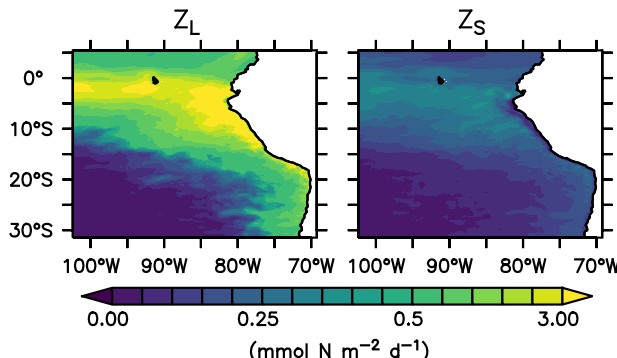

**Figure 3.** Nitrogen flux from large (left) and small (right) zooplankton to detritus due to zooplankton mortality, integrated over the upper 100 m of the water column (mmol N m$^{-2}$ d$^{-1}$)

1) in the reference scenario is 3.1 mmol N m$^{-2}$ d$^{-1}$ ($\mu_{Z_L} \cdot [Z_L]^2$, Figure 3). Neglecting any non-linear and feedback effects within the model, a 50 % change of mortality rate would result in a change of zooplankton loss due to mortality of 1.55 mmol N m$^{-2}$ d$^{-1}$. It is thus a conservative value, compared to a change of 5 mmol N m$^{-2}$ d$^{-1}$ that anchovy population fluctuations could theoretically exert, as estimated in Sect. 1.

All model experiments and the reference scenario were spun up for 30 climatological years. Annual means of the state
variables and nitrogen fluxes from the last climatological year of the high resolution domain were analysed.

## 2.5  Model analysis

The ETSP is highly dynamic at temporal and spatial scales, with nutrient-rich cold water near the coast of Peru, and oligotrophic regions offshore. Therefore, we analysed four different regions: the "full domain" without boundaries (F), an "oligotrophic region" offshore (O), and the "coastal upwelling" (C) section near the Peruvian coast (Figure 1). For most of our analysis,
percentage relative differences between the reference scenario and the other scenarios were calculated. In addition, we analyse the development of a plankton succession from the coast of Peru towards the open ocean at 12° S (Figure 1, white line). All analyses in our study take into account only annual averages. However, we recognise that there is a high temporal variability in the NHCS (see Appendix C).

## 3  Results

We first provide an overview over the general performance of the reference scenario, with respect to the different model components and biogeochemical provinces (Section 3.1). We then investigate their response to changes in zooplankton mortality, and the response of the plankton ecosystem structure (Section 3.2). The coastal upwelling region (C) is especially productive and habitat of the largest aggregations of small pelagic fish, whose temporal variability inspired this study. Therefore, in Sect. 3.3 we place special emphasis on this region. Finally, we investigate the response of the zooplankton losses due to mortality in





the experiments, in order to understand whether the model structure buffers or increases the effect of varying the zooplankton mortality rate on such term (Section 3.4). This would give us an insight into potential feedbacks to higher trophic levels.

### 3.1 Biogeochemistry and plankton distribution in the reference scenario

In the reference scenario (Figure 4, rows 1, 6 and 11), oxygen concentration increases offshore, with average values in the oligotrophic region (O) of 226.6 at the surface and 146.6 mmol $O_2$ m$^{-3}$ in the deep water (below 100 m). On the other hand, in
the Oxygen Minimum Zone (OMZ) under the highly productive coastal upwelling region (C), oxygen becomes as low as 5.3 mmol $O_2$ m$^{-3}$. As a result of denitrification, nitrite concentrations are high in this region, averaging 6.6 mmol N m$^{-3}$. Nitrate is the most abundant nutrient all over the domain. Its deep water concentrations range from 40.4 in C to 34.2 mmol N m$^{-3}$ in O. In contrast, ammonium values are generally lower at depth than in shallow waters, around 0.004 mmol N m$^{-3}$. An exception is C with 0.13 mmol N m$^{-3}$ of ammonium in the deep layer (Figure 4). Please refer to Jose et al. (2017) for a further in-depth
analysis of biogeochemical tracers in the reference scenario.

  Detritus and dissolved organic nitrogen (DON) concentrations are highest at the surface, the exception being large detritus in C, where its concentration increases in deep waters. This inverse distribution of large detritus can be attributed to its high sinking speed of 40 m d$^{-1}$, which rapidly transports organic matter produced in the productive euphotic zone to depth, where it accumulates (because of the model's closed lower boundary; Gutknecht et al., 2013a), and reaches a concentration as high
as 0.22 mmol m$^{-3}$ (Figure 4).

  Phyto- and zooplankton are generally absent in the deep water. In the surface layer, phytoplankton is clearly favoured by nutrient rich coastal upwelling, where total phytoplankton reaches 0.93 mmol N m$^{-3}$ on average, compared to 0.25 mmol N m$^{-3}$ in the oligotrophic region (Figure 4). When zooming into this region (Figure 5) close to the coast, large phytoplankton exhibits a sharp peak which drops offshore. Moving further offshore, large zooplankton peaks at the decline of the large phytoplankton
peak, followed by increased concentrations of small phytoplankton. Given the (Ekman-driven) transport of surface waters (Figure 1) this spatial pattern might be interpreted as a form of succession as the water is advected offshore. In general, modelled concentrations of large zooplankton are high not only in the coastal upwelling region (Section 2.3), but in large parts of the domain (Figure 4), except for the oligotrophic south-western region (Figure 4).

  The spatial pattern of plankton near the coast can be explained by the competitive advantage of large phytoplankton in deep
water due to its steeper initial slope of the P-I curve (see Appendix A, Table A1), eutrophic conditions in the nutrient-rich upwelling water, and relatively low predation due to the lack of large zooplankton. This opens a loophole for large phytoplankton to grow in the upwelling waters. As water is transported offshore (Figure 1), large zooplankton starts to grow and grazes on large phytoplankton. More oligotrophic sunlit conditions even further offshore favour small phytoplankton growth at the surface. Therefore, this first analysis reveals a spatial segregation and succession from the coast to offshore waters. These
patterns are caused by the model's parameterisation of plankton groups, and their mutual interactions.





## 3.2 Response to zooplankton mortality

When changing zooplankton mortality, the inorganic variables (Figure 4, rows 2 to 5) are not noticeably affected by the experiments. As an exception, averaged oxygen concentration between 100 and 1000 m in the coastal region drops in experiments A in both the high (from 5.3 to 1.9 mmol $O_2$ m$^{-3}$) and low (to 2.3 mmol $O_2$ m$^{-3}$) mortality scenarios. This is caused by increased deep large detritus and its subsequent remineralisation in both scenarios. In experiment A_high, deep water DON also increases by 0.03 mmol N m$^{-3}$; the resulting increased remineralisation of this component contributes to the drop in oxygen concentrations mentioned above.

In deep water (between 100 and 1000 m) most of the plankton compartments (Figure 4) present mild to strong relative responses. However, note that concentrations at these depths are generally low; and in fact, the absolute responses are negligible. Because the largest changes occur in plankton stocks at the surface, in the following we focus on the ecological interactions within the upper 100 m, and their response to changes in zooplankton mortality.

Surface plankton responds very similarly to changes in mortality in experiments A and B (Figure 4 and Appendix C, Figure D1). Phytoplankton and large zooplankton follow the same direction of response in all regions: Concentrations of large zooplankton decrease in the high mortality scenario and increase in the low mortality scenario, as could be expected. Large phytoplankton responds inversely to large zooplankton, evidencing a top-down control of its main grazer. On the other hand, the response of small phytoplankton is inverse to that of large phytoplankton. In contrast, small zooplankton shows an asymetric response to changes in mortality, as it mainly decreases in the low mortality scenarios, but responds only weakly in the high mortality scenarios (Figure 4 and Appendix C, Figure D1).

The spatial plankton distribution along the transect (Figure 5) remains the same when zooplankton mortality changes, but the absolute concentrations of each compartment change. In all scenarios large phytoplankton peaks close to the coast. When large zooplankton concentrations are reduced because of its higher mortality in experiment B_high, the large phytoplankton peak increases (Figure 5, top right). Similarly, large phytoplankton decreases with lower zooplankton mortality, due to higher grazing of zooplankton on phytoplankton (Figure 5, top left). This pattern is also similar in experiments A. Because the largest effects occur in the very productive coastal region (C), in the following section we will narrow our analysis to this domain.

## 3.3 Effects on the food web in the coastal domain

An increase in zooplankton mortality causes only small changes in total primary production in the coastal upwelling (C) , but the partitioning between the two phytoplankton groups changes (Figure 6). In particular, total primary productivity of the system is increased by 3.9 % in B_high and reduced by 5.5 % in B_low. Large phytoplankton is the dominant group. However, its productivity increases by about 19 % in B_high and decreases by 22 % in B_low, i.e. its changes are much more pronounced than the overall phytoplankton response. Because small phytoplankton shows an inverse response in production, this dampens the change in total primary production. Thus, a low zooplankton mortality favours small phytoplankton and its growth, and a high mortality favours large phytoplankton; changes in both phytoplankton groups result in a weak response of total primary production.





Experiment B_high exhibits the highest total plankton biomass in the upper 100 m of the upwelling system (112.6 mmol N m$^{-3}$,

Figure 6), which is mostly concentrated in the large phytoplankton compartment (59.43 mmol N m$^{-3}$). In this experiment the main pathway of nitrogen transfer to large zooplankton is via its grazing on large phytoplankton (6.77 mmol N m$^{-3}$ d$^{-1}$). As mortality decreases, small phytoplankton and large zooplankton gain biomass. Large phytoplankton grazing remains the main nitrogen source for large zooplankton. However, large zooplankton consumption of small phytoplankton is almost 3 times higher in B_low than in B_high (Figure 6). Thus, a reduction in mortality causes a switch in the diet of large zooplankton, from

mainly large phytoplankton to a diet that consists of more than one quarter of small phytoplankton.

Small zooplankton biomass decreases by ≈0.4 mmol N m$^{-3}$ (about 5 % of the reference value) in B_low but it only increases by 0.15 mmol N m$^{-3}$ (about 2 %) in B_high (Figure 6). Despite the changes in small zooplankton biomass, the consumption of its biomass by large zooplankton remains approximately the same in all experiments, resulting in a higher proportional biomass loss of small zooplankton in scenario B_low. Hence, predation by large zooplankton as well as competition for food negatively

affect small zooplankton. Under high mortality conditions, the availability of large phytoplankton as food increases (Figure 6). However, small phytoplankton, the preferred prey of small zooplankton, declines as explained above. Such antagonistic effects on small zooplankton buffer its response in this scenario.

### 3.4    Zooplankton losses due to mortality response

The nitrogen loss due to mortality (term $\mu_{Z_i} \cdot [Z_i]^2$ of Eq. (4) and (5)) in experiment B exhibits a different response in large

and small zooplankton. For the coastal upwelling region, zooplankton mass loss due to mortality increases (decreases) for both zooplankton groups in B_high (B_low) (Figure 6). However, the response relative to the reference scenario is mild for large zooplankton, and fluctuates between ±30 % outside the oligotrophic area (Figure 7). In contrast, small zooplankton loss due to mortality exhibits a clear relative increase all over the domain in B_high, and a decrease in B_low. The moderate response of large zooplankton loss can be attributed to the combined effects of changes in concentration and changes in $\mu_{Z_L}$: Large

zooplankton concentration increases when $\mu_{Z_L}$ is decreased, and vice versa. This opposite trend buffers the effect of a change in the mortality rate. On the other hand, small zooplankton concentration changes in the same direction as $\mu_{Z_S}$, due to the combined effects of changes in its concentration, due to grazing pressure exerted by large zooplankton and competition for food with this group, and changes in $\mu_{Z_S}$.





**Figure 4.** Annually, spatially (oligotrophic (O) region, full domain (F) and coastal upwelling (C) region; see guide and Figure 1 for further reference) and depth (0-100 m and 100-1000 m) averaged concentrations (mmol N m$^{-3}$ or mmol O$_2$ m$^{-3}$) of the biogeochemical prognostic variables in the model reference scenario (rows 1, 6 and 11); and normalised percent difference between the reference scenario and experiments. "P" stands for phytoplankton, "Z" for zooplankton, "D" for detritus, "DON" for dissolved inorganic nitrogen, "T" for total, "L" for large and "S" for small.





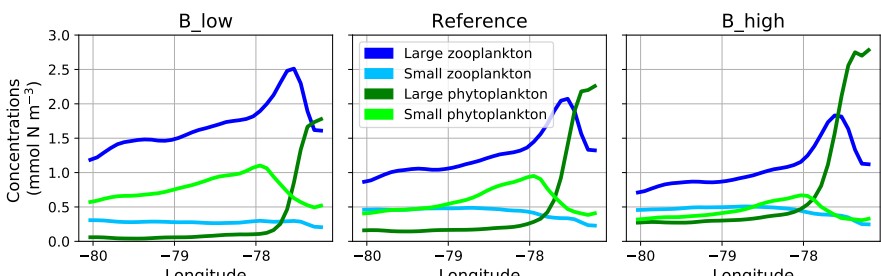

**Figure 5.** Zonal distribution of surface plankton concentrations at 12° S annually averaged in the reference and the two B scenarios, respectively.





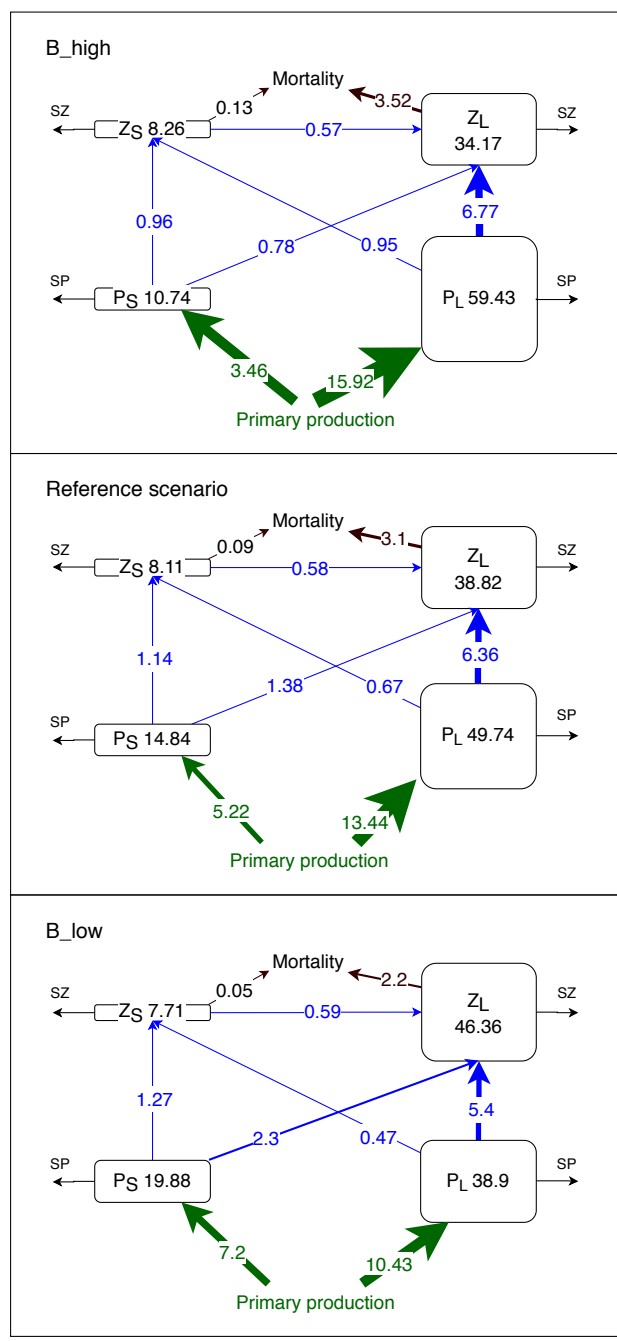

**Figure 6.** Concentrations (mmol N m$^{-2}$) and nitrogen fluxes (mmol N m$^{-2}$ d$^{-1}$) between plankton compartments (small and large phytoplankton $P_S$ and $P_L$, and small and large zooplankton $Z_S$ and $Z_L$, respectively) integrated over the upper 100 m, and averaged over latitude and longitude in the coastal upwelling region (see Figure 1). SZ and SP indicate the sinks which include: phytoplankton mortality, zooplankton metabolism, large phytoplankton sedimentation, unassimilated primary production and unassimilated grazing (Gutknecht et al., 2013a).

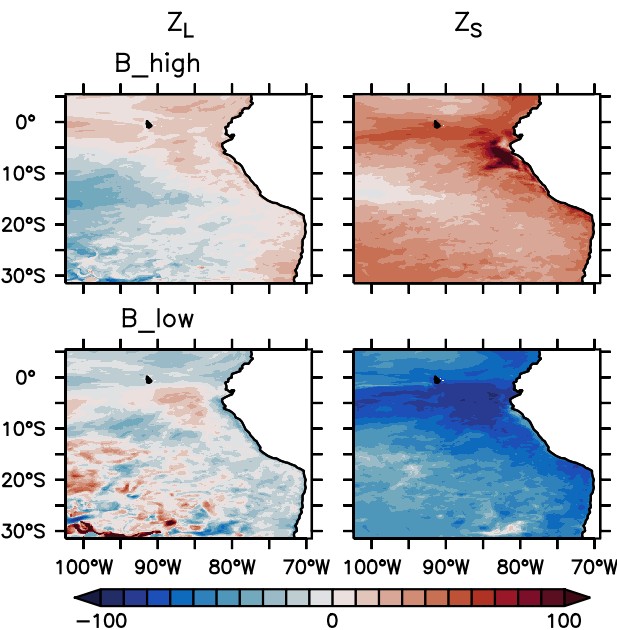

**Figure 7.** Percentage normalised difference in the nitrogen flux from large and small zooplankton to detritus due to zooplankton mortality, integrated over the upper 100 m of the water column, between experiment B_high and experiment B_low, and the reference scenario (see Figure 3 for the reference scenario).



To summarise, increasing and decreasing zooplankton mortality by 50 % generates a rearrangement of the plankton ecosys-
tem; however, the overall changes in the large zooplankton loss are not as high as it would be expected from a change in
mortality rate alone. This might buffer the system once the biogeochemical model is coupled to a model of higher trophic
levels.

## 4 Discussion

### 4.1 Constraining the zooplankton compartment

An increasing need for the development of end-to-end models has brought interest into using results of biogeochemical models
as forcing for higher trophic level models (fish, macroinvertebrates and apex predators) (see Tittensor et al., 2018, for a review).
In a one-way coupling set-up, the biomass of plankton available as food for higher trophic levels has been adjusted during cal-
ibration of the latter, reducing the amount of plankton that is available for fish consumption (e.g., Oliveros-Ramos et al., 2017;
Travers-Trolet et al., 2014b). However, for two-way coupling set-ups, this adjustment of the available plankton biomass could
buffer the effect of higher trophic levels on lower trophic levels (e.g., Travers-Trolet et al., 2014b). Biogeochemical models
can produce a wide range of output depending on their parameter values (Baklouti et al., 2006), and their non-linearity (Lima
et al., 2002). Few studies have aimed to understand such behaviour (Baklouti et al., 2006) and examined the sensitivity of the
model to parameters (Arhonditsis and Brett, 2004; Shimoda and Arhonditsis, 2016). Our model study was partly motivated
by the uncertainty associated with the zooplankton mortality. Indeed our model showed that a small alteration in the mortality
parameter (small compared to the wide range of values that have been used for this parameter in different biogeochemical
models) can strongly affect the mass flux within the simulated ecosystem. Hence, there is an increasing need for an accu-
rate plankton representation in biogeochemical models without dismissing other compartments, such as nutrients or oxygen.
Nevertheless, lack of data for validation especially of higher trophic levels is a common problem for biogeochemical models
of the Northern Humboldt Upwelling System (Chavez et al., 2008). Oxygen, chlorophyll and nitrate in our model have been
evaluated previously (Jose et al., 2017). Here we presented the first attempt to compare the large zooplankton compartment of
the ROMS-BioEBUS ETSP configuration with mesozooplankton observations.

At the surface, zooplankton concentrations simulated by our model in the reference scenario are 1 order of magnitude higher
than observations at most stations. However, sampling in the upper 10 m depth may be impacted by water disturbance by the
ship adding additional errors to the measurements. The match to observations improves with depth. Modifying the mortality
rate by +50 % (-50 %) produced only a change of -12 % (19 %) in large zooplankton concentration, indicating that either the
induced changes in mortality rate were not large enough, or that this parameter is not overly influential to improve the model fit
to observations. For an average nitrogen flux due to large zooplankton mortality of 2.2, 3.1 and 3.52 mmol N m$^{-2}$ d$^{-1}$ and ZL
integrated concentrations of 46.36, 38.82, 34.17 mmol N m$^{-2}$ (see Section 3.3), the mortality rate would be 0.04, 0.08 and 0.1
d$^{-1}$ in scenarios B_low, reference and B_high respectively. In all cases the values are lower than the 0.19 d$^{-1}$ estimate by Hirst
and Kiørboe (2002) for copepods in the field at 25° C. The closest scenario to observarions is B_high where the mortality rate





is only about half of the estimate by Hirst and Kiørboe (2002). This is also the scenario that better resembles mesozooplankton observations, since it exhibits the lowest concentrations.

Some part of the mismatch between model and observations might be related to how both data types are generated. Therefore, a direct comparison between model and observations has to be viewed with some caution. In our model, large zooplankton acts
as a closure term which is adjusted to balance the biomass and nitrogen flux to other compartments, and does not resemble a specific set of species. Its parameters (maximum grazing rate, feeding preferences, etc.) are meant to represent larger, slow-growing, species with a preference for diatoms. As such, they might not be directly comparable to the observed groups. The observations, on the other hand, do not cover the whole taxonomic and size spectrum of mesozooplankton. For instance, no gelatinous organisms are accounted for, and only mesozooplankton greater than 500 $\mu$m is considered in the sampling (Kiko
and Hauss, 2019), and fragile organisms, such as rhizaria, are not quantitatively sampled by nets (Biard et al., 2016). Therefore, the observations might be biased low in comparison to the model.

The spatial variability between different profiles of zooplankton is greater in the observations than in the model, and the variability of concentrations within each single profile is much larger than the differences between the modelled mortality scenarios (Figure 2). This indicates that the model may not include all potential sources of variability. Finally, we only com-
pared our simulated zooplankton against night observations because in our model zooplankton is always active at the surface. In reality, zooplankton is known to perform diel vertical migrations (DVM), which could increase the export flux to the deep ocean (Aumont et al., 2018; Archibald et al., 2019; Kiko and Hauss, 2019; Kiko et al., 2020). The lack of DVM could affect the export of organic matter to greater depths, and therefore the biogeochemical turnover at the surface. Zooplankton likely also experiences lower mortality at depth (Ohman, 1990); however, off Peru these benefits might be counterbalanced by reduced
oxygen availability and the concurrent metabolic costs. These obstacles for comparing zooplankton models with observations have been described by Mullin (1975) already more than four decades ago: a) "the zooplankton is a very heterogeneous group, defined operationally by the gear used for capture rather than by a discrete position in the food web" (Mullin, 1975). b) zooplankton is irregularly distributed in space, not necessarily following physical features. c) adult stages of some zooplankton groups perform vertical migrations (Mullin, 1975).

To summarise, some biases and mismatches between model and observations remain; given the uncertainties and episodic nature associated with the observations, and their correspondence to their model counterparts, further studies will be necessary to more precisely calibrate the model. For a complete model evaluation, however, also the small zooplankton compartment should be evaluated against microzooplankton samples. The high mortality scenario, B_high, is the one that is closest to the observations, due to producing the smallest concentrations of large zooplankton at the surface. However, changing this
parameter was obviously not enough to match the observations. In fact, in our model an increase of 50 % in the mortality rate produced only an increase of 14 % (0.4 mmol N m$^{-2}$ d$^{-1}$) in large zooplankton mortality loss (see Section 3) because of the high non-linearity of the model. Indeed, potential changes in zooplankton losses of 5 mmol N m$^{-2}$ d$^{-1}$, derived by fluctuations in anchovy stocks and grazing pressure (see Section 1) point towards much larger values for the mortality rate. An even stronger increase in zooplankton mortality rate (e.g., Lima and Doney (2004) applied a 5 times larger value), along
with a subsequent adjustment of other parameters may be necessary to approach observed values. In addition, complementary





observations with other sampling methods, could provide a better estimation of mesozooplankton concentrations for tuning the model.

## 4.2 Zooplankton mortality and the response of the pelagic ecosystem

Our model study showed the strongest response of the ecosystem to changes in mortality rate in the highly productive coastal

upwelling. Here, the response of the model ecosystem was mainly driven by large zooplankton. This can be concluded from the close similarity of model solutions A_high and B_high, as well as A_low and B_low, respectively (see Appendix D). The mortality term for small zooplankton played a lesser role; in addition to the direct effect of mortality rate, this compartment was also affected by grazing by, and competition with, large zooplankton. Large zooplankton fluctuations due to mortality directly affected large phytoplankton through grazing. Small phytoplankton, on the other hand, was affected by grazing but also by

competition with large phytoplankton. Changing the mortality rate produced little effect on the mass loss of large zooplankton due to mortality; however, it altered the nitrogen pathways along the trophic chain, and ultimately the concentrations of most plankton groups, yet in different ways, depending on the direction of change. Under conditions of high zooplankton mortality the food chain is dominated by nitrogen transfer from large phytoplankton to large zooplankton, the classical foodweb attributed to highly productive upwelling systems (Ryther, 1969). When zooplankton mortality is reduced, large zooplankton increases

its consumption of small phytoplankton, taking over the role of small zooplankton.

In our model, large zooplankton has a competitive advantage by feeding on its competitor, small zooplankton, a strategy that was also found to evolve in simple ecosystem models as an advantageous alternative to direct competition (Cropp and Norbury, 2020). We find that under low mortality conditions, this advantage increases. The importance of competition is further evidenced in the changes in small phytoplankton concentrations in the coastal upwelling region. These were partly driven by

changes in the availability of resources arising from fluctuations in large phytoplankton concentrations, which constitute the dominant group in the coastal upwelling. Natural selection, competitive exclusion and different resources utilisation strategies, together with bottom-up forcing by the physical processes in the environment, can shape the plankton community in global models (Follows et al., 2007; Dutkiewicz et al., 2009; Barton et al., 2010), and indicate bottom-up effects on the phytoplankon community. On the other hand, Prowe et al. (2012) showed that variable zooplankton predation can increase phytoplankton

diversity by opening refuges for less competitive phytoplankton groups, and thus exert top-down effects. In our study, the biological interactions between two phytoplankton groups, mainly competition for resources (bottom-up), are additionally affected in a top-down manner by changes in zooplankton concentrations.

The processes driving the ecosystem response in our regional study are dominated by trophic interactions among the size classes of phytoplankton and zooplankton. We found a top-down driven response affecting mainly the plankton compartments

of the model. The direction of the total zooplankton and total phytoplankton change is determined by the large zoo- and phytoplankton groups. Small zoo- and phytoplankton buffer the response when they present opposite trends to their larger counterparts (Table 1). Overall, total zooplankton decreases (increases) by -11 % (10 %), and total phytoplankton changes by about +(-)6 % in the high (low) mortality scenario (Figure 4 and Table 1). In the study by Getzlaff and Oschlies (2017), the main driver is also zooplankton but the response depends only on one zooplankton size class. Zooplankton biomass changes



by about -(+)5 %, and phytoplankton biomass by about +(-)1 % in the high (low) mortality scenario (Getzlaff and Oschlies, 2017, Figure 2). Hence, the effects of changing zooplankton mortality follow the same trend and are slightly milder than in our study.

At the regional scale, the responses in Getzlaff and Oschlies (2017) study have a feedback from lower trophic levels, either to phytoplankton in the nutrient repleted region, or all the way to nutrients in the oligotrophic region. Therefore, the biomass of both zooplankton and phytoplankton is ultimately bottom up affected. This is evidenced by a change in zooplankton and phytoplankton concentrations in the same direction (Getzlaff and Oschlies, 2017, Figure 3). On the other hand, although our study also exhibits a feedback from phytoplankton to zooplankton, the strongest driver remains the top-down predation of zooplankton on phytoplankton (Table 1).

The regional differences between our study and that by Getzlaff and Oschlies (2017) can likely be explained by the different model structures and experimental setups, namely the number of phyto- and zooplankton compartments, different time scales considered for model simulation and analysis, and the spatial domain: While Getzlaff and Oschlies (2017) applied a global biogeochemical model with just one size class of phyto- and zooplankton, simulated until near steady state; our regional model study applies a more complex biogeochemical model, simulated for only 30 years. Further, the short few-year timescale of our model simulations might have prevented the effects of changed zooplankton mortality from stabilising at the surface and propagating to deeper water layers which contain the largest concentrations of nutrients. Finally, the region modelled in our study is spatially very dynamic already at the mesoscale resolution, as evidenced by a well-defined plankton spatial succession from the coast of the continent towards the open ocean.

The phytoplankton bloom, which develops during the transition from coastal areas towards the open ocean can be explained by an imbalance between sources and sinks, triggered by changing environmental conditions. For example, Irigoien et al. (2005) applied the concept of 'loopholes' proposed by Bakun and Broad (2003) to explain fish productivity and recruitment success, to phytoplankton: according to their concept, phytoplankton blooms are formed when environmental conditions open a loophole in the plankton community of a mature ecosystem. Then, particularly phytoplankton species that are able to escape microzooplankton predation are those that will take advantage of the loophole and bloom (Irigoien et al., 2005). Our model results suggest that similar processes occur. Low concentrations of large zooplankton allow large phytoplankton to bloom near the coast. While the water is advected offshore, zooplankton growth and grazing offset the bloom. Observations by Franz et al. (2012) also reported a spatial succession with large diatoms abundant in the coastal upwelling region being replenished by nanophytoplankton offshore. However, they propose silicate as the limiting nutrient offsetting the diatoms peak, which is not present in our model. On the other hand, the global model by Getzlaff and Oschlies (2017) does not resolve mesoscale processes. Furthermore, while they divided their study in tropics, as an oligotrophic region, and Southern Ocean, as an upwelling region, the upwelling system off Peru in the Eastern Tropical South Pacific is a nutrient rich area. For all of this, we based our comparison on similarities in the nutrient concentration (high nutrients, oligotrophic and whole domain), rather than on geographic overlap.

Both studies have a quadratic zooplankton mortality term (see Section 2.2.2), and changed the zooplankton mortality parameter by ±50 %. In Getzlaff and Oschlies (2017) study, the zooplankton loss due to mortality is not provided. However, it can be





**Table 1.** Qualitative comparison of the response of total, large and small zooplankton ($Z_T$, $Z_L$, $Z_S$) and phytoplankton ($P_T$, $P_L$, $P_S$) to 50 % higher and lower zooplankton mortality parameter in our experiments B, with the results from Getzlaff and Oschlies (2017) ($Z_{GO}$, $P_{GO}$) Full, oligotrophic and coastal upwelling refer to the regions in our study (see Figure 1) integrated over the upper 100 m. Global, Southern Ocean and Tropics refer to the study by Getzlaff and Oschlies (2017, Figures 2 and 3 at year 300).

| | $Z_T$ | $P_T$ | $Z_L$ | $P_L$ | $Z_S$ | $P_S$ | $Z_{GO}$ | $P_{GO}$ |
|---|---|---|---|---|---|---|---|---|
| **Full/ Global** | | | | | | | | |
| High | - | + | - | + | - | - | - | + |
| Low | + | - | + | - | - | + | + | - |
| **Oligotrophic/ Tropics** | | | | | | | | |
| High | - | + | - | + | - | + | - | - |
| Low | + | - | + | - | + | + | + | + |
| **Coastal Upwelling/ Southern Ocean** | | | | | | | | |
| High | - | + | - | + | + | - | + | + |
| Low | + | - | + | - | - | + | - | - |

calculated from the zooplankton concentration and mortality rate. Assuming an integrated zooplankton biomasses at year 300 (Getzlaff and Oschlies, 2017, Figure 2) of 98, 93 and 89 Tg N in the low, reference and high mortality scenarios respectively; and mortality parameters of 0.03, 0.06 and 0.09 (mmol N m$^3$)$^{-1}$ d$^{-1}$; then there is a difference of -44.5 % and 37.4 % in the zooplankton loss due to mortality between the low and reference scenario, and the high and reference scenario respectively. As shown in Sect. 3.4, the mortality rate in our study is also smaller, than the ±50 % changes that would be expected from a

change in the mortality parameter of ±50 % (see Section 2.4). The non-linearity of both global and regional models seems to reduce the effect of changes in the mortality parameter on zooplankton loss.

In summary both studies show a similar global response to changes in zooplankton mortality, driven by zooplankton preying on phytoplankton. Two zooplankton and phytoplankton size classes present opposite trends in our studies, buffering the overall response. Nevertheless, the relative changes in total zooplankton and total phytoplankton are in the same order of magnitude

as in Getzlaff and Oschlies (2017) study and even slightly higher. Regionally different feedbacks operated in the two models, possibly due to the specific set up of each study, spin up time, and resolution. Finally, the relative change in the zooplankton loss due to mortality is smaller than the expected ±50 % in both studies.

## 4.3 From plankton to higher trophic levels

In our study, we changed the zooplankton quadratic mortality, which could be regarded as the effect of a predator targeting

highly aggregated zooplankton populations, or whose concentration closely follows that of zooplankton. This can be viewed as a way to parameterise the effect of changing fish abundance on the biogeochemistry of the system. In this case, a low



zooplankton mortality implies less small pelagic fish (such as anchovies and sardines), while a high zooplankton mortality implies a higher abundance of such fish. Further, our experiments are based on two different assumptions: one where small pelagic fish feed only on large zooplankton (experiments A), and one where they feed and affects the mortality of both large and small zooplankton (experiments B).


The diet of anchovy is still under debate. While previous studies had considered that anchovies feed mainly on phytoplankton, Espinoza and Bertrand (2008) concluded that anchovies feed on zooplankton, especially euphausiids and copepods. Furthermore, the diet of anchovy seems to be more flexible than previously considered (Espinoza and Bertrand, 2008). For instance, the anchovy collapse in 1972 was correlated with a shift from a population feeding mostly on phytoplankton to a southern population feeding on zooplankton (Hutchings, 1992). On the other hand, small zooplankton groups such as ciliates

have been reported as a minor component of anchovies diet (Espinoza and Bertrand, 2008, Table 5). Thus, experiments A are more likely to resemble the fluctuations in anchovy populations. On the other hand, sardines, with their finer gill rakers, obtain most of their nutrition from microzooplankton (van der Lingen et al., 2006). Although currently sardines are not as abundant as anchovies off Peru, historically they also did build up large concentrations and strong fluctuations over time were observed (Lluch-Belda et al., 1989; Rykaczewski and Checkley, 2008). Thus, when considering also sardine populations and feeding

modes, experiments B (simultaneous mortality change in both large and small zooplankton) might be more appropriate to parameterise the effects of changed fishing mortality on lower trophic levels.

Although the quadratic mortality rate is constant over the entire domain, the fractional loss rate by zooplankton mortality ($\mu_{Z_i} \times Z_i$, d$^{-1}$) varies over the domain because of changes in zooplankton concentration. This might mimic spatially variable

grazing pressure by fish. However, our experimental setup might be too simple to investigate the detailed response of predator-prey relationships. We partly tried to avoid too general conclusions by focusing our analysis on the coastal upwelling region off Peru, since anchovies are highly concentrated in this region (Checkley et al., 2009). In addition, we neglected any feedback effects between zooplankton and their predator fish. A more detailed model setup, as, for example, in coupled biogeochemical-fish models (Oliveros-Ramos et al., 2017) would help to elucidate the specific trophic interactions in this region, and their

response to environmental changes, and changes in fishing pressure.

## 5 Conclusions

In summary, our study showed that changes in zooplankton mortality can have a strong impact on the trophic interactions between the plankton compartments of the model. Such changes are meant to mimic variations in the abundance of planktivorous fish in the system. Large zooplankton mortality, as the top predator in the model, is the main driver of the planktonic foodweb

response in the model. Changes in the mortality rate of small zooplankton, which may resemble fluctuations in the sardine populations, are masked when large zooplankton mortality also changes. The high zooplankton mortality scenario, which mimics an increase in planktivorous fish, generates a shorter foodweb where most of the nutrients are taken up by large phytoplankton. In the low mortality scenario, the biomass of small phytoplankton increases and a longer food chain where nitrogen reaches large zooplankton through consumption of small zooplankton is favoured. Our 50 % mortality changes are small compared to





changes expected from the population fluctuations that small pelagic fish have historically experienced in the NHCS (Sections 1 and 2.4). The causes for fluctuations in small pelagic landings are not fully understood, but largely attributed to changes in environmental conditions such as ENSO. In a highly bottom-up driven system, it is important to be cautious and conservative when evaluating top-down scenarios. A fully coupled end-to-end ecosystem model explicitly including fish (as by, for example, Travers-Trolet et al., 2014b), would allow to represent the effect of temporal and spatial variability of fish. It would also allow

for a specialised targeting of fish food, and for including the bottom-up effect of changing zooplankton concentration on fish populations, as well as their top-down effect, and its potential consequences for the entire ecosystem. However, it would also involve the inclusion of more parameters in the model (up to hundreds of parameters, see Oliveros-Ramos et al., 2017), which are only poorly constrained. Therefore, while explicitly including fish in a model widens the possibilities for controlling the system, it may also increase the sources of uncertainty. Here we utilised an already validated physical and biogeochemical

model, and parameterised the loss of zooplankton due to fluctuations in small pelagic fish, without adding additional complexity to the model. Our results may be a baseline reference for further studies exploring such effect.

*Code and data availability.* Code and data used in this study are available upon request. The ROMS model is maintained at: https://www. myroms.org.

## Appendix A: Plankton parameters

Our reference simulation has the same configuration as in Jose et al. (2017) which in turn is based on the parameters used in Gutknecht et al. (2013a), with minor adjustments. Tab. A1 provides a list of the most relevant parameters for this study, as well as its description and units. This is not a comprehensive list, for the full list of parameters and its values please see Gutknecht et al. (2013a); Jose et al. (2017).

## Appendix B: Mesozooplankton evaluation

Modelled large zooplankton is higher than observed nighttime mesozooplankton above 100 m (Figure B1). Deep water zooplankton is absent in the model, while observed mesozooplankton is present at depths between 600 and 1000 m. Mesozooplankton below 200 m further increases during daytime (data not show, please see Fig. 4 in Kiko and Hauss (2019)).

## Appendix C: Temporal variability

The NHCS presents high climatological and interannual variability. In our modelled region, small phytoplankton concentrations

are relatively stable throughout the year. On the other hand, large phytoplankton production exhibits a clear seasonal pattern, with the largest concentrations presented in austral summer (Figure C1). Echevin et al. (2008) discuss a similar seasonal pattern found in their model. Such pattern has also been identified in satellite-derived primary production (Messié and Chavez, 2015).





**Table A1.** Plankton parameters in the model. The complete list of biogeochemical parameters is given in Gutknecht et al. (2013a), and the values for the reference simulation are provided in Jose et al. (2017).

| Symbol | Description | Value | Units |
|---|---|---|---|
| $\alpha P_S$ | Initial slope of P-I curve for $P_S$ | 0.25 | $(W\ m^{-2})^{-1}\ d^{-1}$ |
| $\alpha P_L$ | Initial slope of P-I curve for $P_L$ | 0.04 | $(W\ m^{-2})^{-1}\ d^{-1}$ |
| $\mu_{P_L}$ | Mortality rate of $P_S$ | 0.027 | $d^{-1}$ |
| $\mu_{P_L}$ | Mortality rate of $P_L$ | 0.03 | $d^{-1}$ |
| $K_{NH4_{P_S}}$ | Half-saturation constant for uptake of $NH_4$ by $P_S$ | 0.5 | $mmol\ N\ m^{-3}$ |
| $K_{NH4_{P_L}}$ | Half-saturation constant for uptake of $NH_4$ by $P_L$ | 0.7 | $mmol\ N\ m^{-3}$ |
| $K_{NO3_{P_S}}$ | Half-saturation constant for uptake of $NO_3 + NO_2$ by $P_S$ | 1.0 | $mmol\ N\ m^{-3}$ |
| $K_{NO3_{P_L}}$ | Half-saturation constant for uptake of $NO_3 + NO_2$ by $P_L$ | 2 | $mmol\ N\ m^{-3}$ |
| $w_{P_L}$ | Sedimentation velocity of $P_L$ | 0.5 | $m\ d^{-1}$ |
| $f1_{Z_S}$ | Assimilation efficiency of $Z_S$ | 0.75 | – |
| $f1_{Z_L}$ | Assimilation efficiency of $Z_L$ | 0.7 | – |
| $g_{max_{Z_S}}$ | Maximum grazing rate of $Z_S$ | 0.9 | $d^{-1}$ |
| $g_{max_{Z_L}}$ | Maximum grazing rate of $Z_L$ | 1.2 | $d^{-1}$ |
| $e_{Z_S P_S}$ | Preference of $Z_S$ for $P_S$ | 0.75 | – |
| $e_{Z_S P_L}$ | Preference of $Z_S$ for $P_L$ | 0.25 | – |
| $e_{Z_L P_S}$ | Preference of $Z_L$ for $P_S$ | 0.26 | – |
| $e_{Z_L P_L}$ | Preference of $Z_L$ for $P_L$ | 0.5 | – |
| $e_{Z_L Z_S}$ | Preference of $Z_L$ for $Z_S$ | 0.24 | – |
| $k_{Z_S}$ | Half-saturation constant for ingestion by $Z_S$ | 1 | $mmol\ N\ m^{-3}$ |
| $k_{Z_L}$ | Half-saturation constant for ingestion by $Z_L$ | 2 | $mmol\ N\ m^{-3}$ |
| $\mu_{Z_S}$ | Mortality rate of $Z_S$ | 0.025 | $(mmol\ N\ m^{-3})^{-1}\ d^{-1}$ |
| $\mu_{Z_L}$ | Mortality rate of $Z_L$ | 0.05 | $(mmol\ N\ m^{-3})^{-1}\ d^{-1}$ |
| $\gamma_{Z_S}$ | Metabolic rate of $Z_S$ | 0.05 | $d^{-1}$ |
| $\gamma_{Z_S}$ | Metabolic rate of $Z_L$ | 0.05 | $d^{-1}$ |

## Appendix D: Plankton surface concentrations

Experiments A and B exhibit very similar spatial trends. Fig. D1 provides an overview of surface plankton concentrations in
their reference scenario, and their changes in the experiments. Note that in the two sets of experiments, large zooplankton increased when mortality decreased and vice-versa. On the other hand, small zooplankton presents a counter-intuitive response when mortality decreases, responding to the change in the concentration of its predator rather than to changes in mortality, and





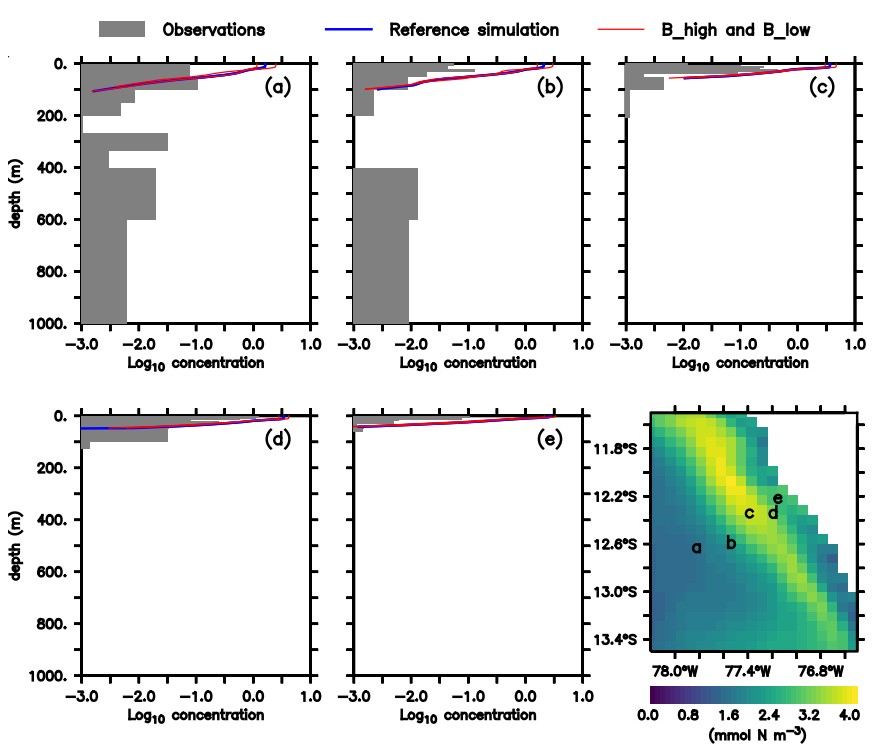

**Figure B1.** Same as Fig. 2 but a) to e) show a depth range up to 1000 m and a logarithmic horizontal axis.

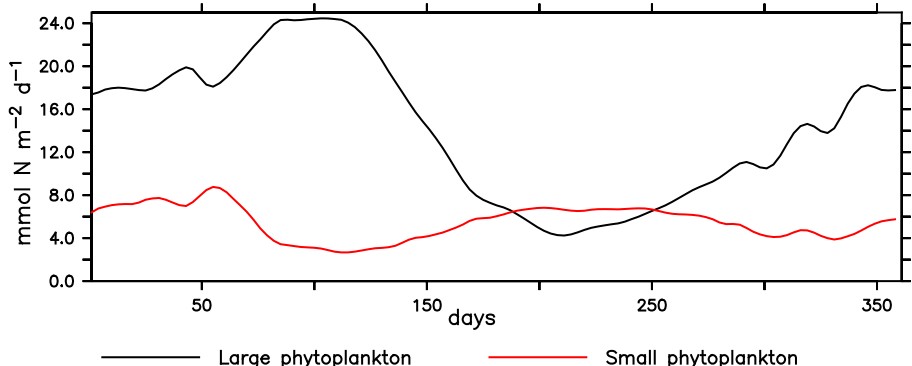

**Figure C1.** Primary production by large and small phytoplankton during the last climatological year of the simulation, averaged over the upwelling region (see Figure 1) and integrated over the upper 100 m in the reference scenario.

an ambiguous response in the high mortality cases. Large and small phytoplankton exhibit opposite trends, seemingly driven by the concentration change of their main predator (large and small zooplankton respectively).



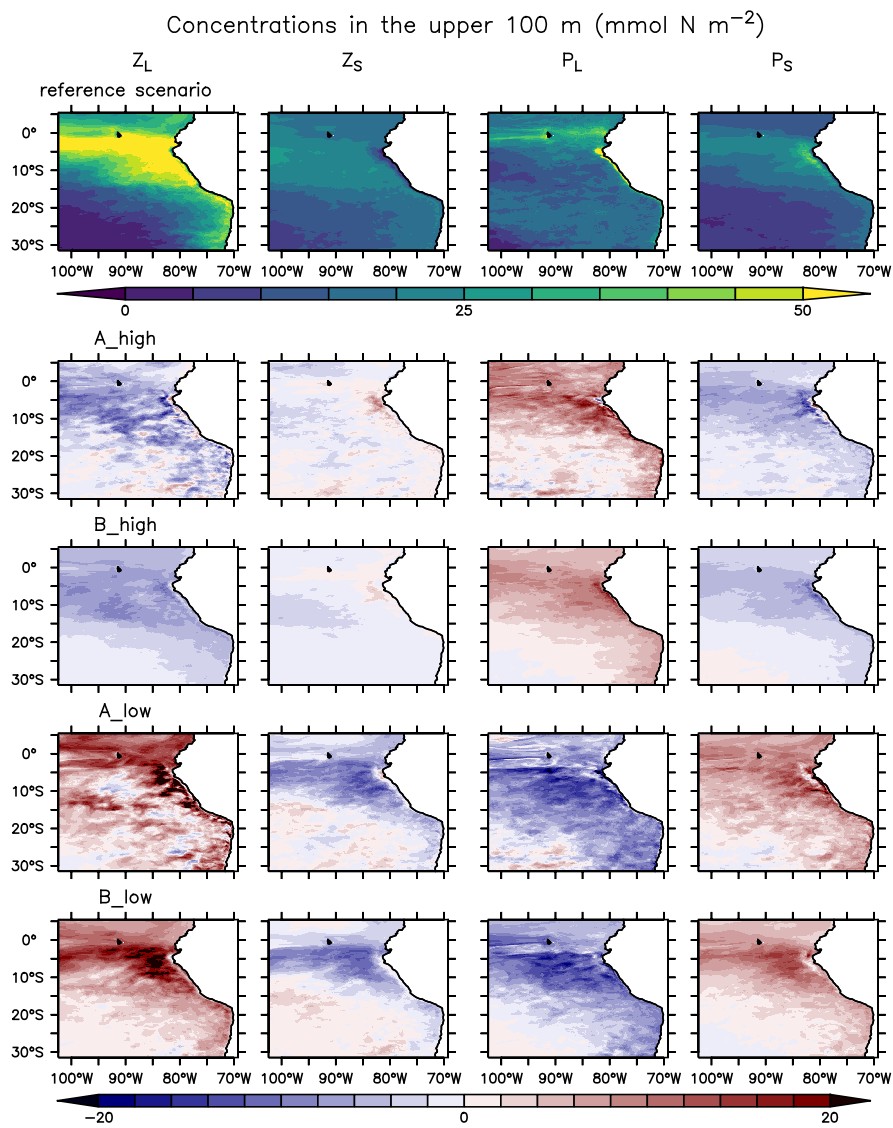

**Figure D1.** Large and small zooplankton ($Z_S$ and $Z_L$) and large and small phytoplankton ($P_L$ and $P_S$) integrated over the upper 100 m of the water column (mmol N m$^{-2}$). Rows from top to bottom: reference scenario, difference between experiments A_high, B_high, A_low, B_low, and the reference scenario.

*Author contributions.* IK and AO designed the study. YSJ carried out the simulations. MHC performed the analysis. RK and HH provided zooplankton observations and expertise on zooplankton dynamics in the NHCS. All authors discussed the results and wrote the manuscript.

*Competing interests.* The authors declare that they have no conflict of interest.





*Acknowledgements.* This is a contribution to the BMBF funded project "CUSCO" Workpackages 5 (RK, HH) and 6 (MHC, IK, AO). Further support for this work was provided by the SFB754. RK furthermore acknowledges support via a "Make Our Planet Great Again" grant of the French National Research Agency within the "Programme d'Investissements d'Avenir"; reference "ANR-19-MPGA-0012". We would like to thank Tianfei Xue for carrying out some of the model simulations.





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
