# Peer review of "Zooplankton mortality effects on the plankton community of the Northern Humboldt Current System: Sensitivity of a regional biogeochemical model"

_Biogeosciences, 2020_

## Referee Comment (RC1) · Anonymous Referee #1 · 2 Jan 2021

Dear authors,

General comments

The manuscript describes the responses of the ETSP region in terms of model experiments by changing mortality rates of the two zooplankton compartments that the model includes. It is an interesting study and it is related to the scope of the journal.

Overall, I find the model does not match very well the observations at surface level (approx. 20m depth). This is indeed discussed in the manuscript and special attention

should be put in these differences, more details should be provided about the sources for these differences. The main strength of this manuscript is represented by the analysis of the experimental design and the responses of the different compartments in the model and how they relate to the region chosen in the domain. I think this manuscript is within the standards of excellence of this journal in terms of scientific quality and significance.

Specific comments

Abstract

My main comment from the abstract and throughout the manuscript is to be consistent with the names you use for the compartments of the model: either use 'large' and 'small' zooplankton or use 'meso' and 'micro' zooplankton. The same comment goes for the compartments of phytoplankton: nano- and micro-phytoplankton or large and small phytoplankton. If they do not mean the same, you should state this, but if they do, you must be consistent with the use of those names.

Introduction

L29 'presumably due to sensitivity to environmental variability' – can you specify what environmental variability you refer to?

L32-38 This paragraph should be rephrased or use a connector when you explain the two species.

L50-51 The last sentence in the paragraph should be rephrased for better clarity as it is difficult to understand.

L58 Can you specify what is the definition of linear and quadratic mortality?

Methods

Figure 1 – in the introduction you mention that anchovy spans from northern Peru to Talcahuano. Can you add landmarks in the map to show where is Peru and Chile? And

why did you choose that domain for each region (O, C, F)?

L108 – 126 When you define the compartments for phytoplankton and zooplankton, it would be good to give a quantitative definition for each compartment. For example, in terms of size class: does microzooplankton include all species that are < 200 $\mu$m? And the same for the other compartments.

L139 – 146 Do you consider temperature dependence in zooplankton processes? How can this affect your results if you would include it? Or justify/discuss why not having it and the effect this could have in your results if it is relevant.

L205 It might be worth briefly describing the way the model calculates the nitrogen and state variables fluxes or at least reference a paper that describes those processes.

Results

L258-259 P_S shows an important difference in deep water for the A_high and B_low experiments, are those differences negligible? It might be good to at least mention or have a brief explanation about those differences

Discussion

L321 Please give some examples about the non-linear processes you refer to in this sentence.

L333-334 Consider not only sampling issues in the differences in Figure 2. What about assumptions in the model? Or bad parameterisation (not only in terms of zooplankton mortality)? Also, you should mention if using a quadratic mortality rate instead of a linear one could produce differences in your results.

L340 It is not clear to me if you just assumed the values given by Hirst & Kiorboe (2002) for you experimental design. Could the values given by Hirst and Kiorboe (2002) be affected by temperature? Is 25degC a realistic value for your domains (see Fig.1). How much change could be expected in terms of zooplankton mortality rates for 2degC

difference? Or is that negligible ?

Conclusions

L497 You mention some effects from ENSO but there is not much discussion associated to this in section 4.

Technical corrections

L28 The name of that city is Talcahuano, not Telcahuano.

L119 correct 'dissolved'

L130 correct the symbol for the exudation fraction for eqs. 2, 3 as they do not match both equations.

L157 – 158 correct the typo of the terms PmathrmS, PmathrmL.

L167 missing a parenthesis ending after Gorsky et al. 2010.
* * *

---

## Referee Comment (RC2) · Anonymous Referee #2 · 11 Jan 2021

GENERAL COMMENTS

This sensitivity study of zooplankton mortality in a regional physical-biogeochemical model of the Eastern Tropical South Pacific provides valuable insights into the responses of ecosystem dynamics and biogeochemistry caused by changes in zooplankton abundance. The relatively simple approach enables disentangling the main drivers of change and their relative strength, showing e.g. that large zooplankton assert a top-down control on both small zooplankton and phytoplankton, causing asymmetrical ecosystem responses.

The paper is well-written, fluent and concise, clearly outlining the methods and assumptions, except for some minor issues addressed in the specific comments. It builds on the conceptual framework of Gezlaff and Oschlies (2017), but applies it to a new study system on a different spatial scale using a different model. The comparison of outcomes from the current regional study to the previous global study is one of the main strengths of this paper.

The model used has been fully described and partially validated in previous papers, with the most relevant equations and parameters repeated in this paper. However, I would like clarifications to some parts as written in the specific comments.

I also have two concerns regarding the zooplankton model formulations. It seems to exclude two important functions for zooplankton dynamics: i) excretion of DIN (ammonium) and ii) temperature-dependence of zooplankton growth/metabolism (see e.g. Tian 2006, Ecol. Model.; Richardson 2008, ICES J. Mar. Sci.). The first might be a mistake in the text (lines 141-143), as DIN excretion is included in the model as described by Gutknecht et al. (2013). If these functions are indeed omitted, I would like the authors to motivate why and discuss if they think this could have major implications for the simulated plankton biomasses and biogeochemistry of the system.

The validation of zooplankton biomass over depth shows that the model is not a very accurate representation of the specific system, especially since simulated surface concentrations are one order of magnitude higher than measured ones. However, the authors provide a thorough discussion of this, and I think the results are still sufficient to support the interpretations and conclusions drawn from the study. I believe the results from this study are a valuable contribution both to the general scientific understanding of real-world plankton dynamics and their effects on the ecosystem, as well as to the development of commonly used biogeochemical ecosystem models.

In summary, I think this paper lives up to the standards of Biogeochemistry and recommend that it should be published after minor revisions.

SPECIFIC COMMENTS AND TECHNICAL CORRECTIONS

Title clearly reflects the contents of the paper.

Abstract provides a concise and complete summary.

Equations 2, 3 and line 131. Please check the symbol for exudation fraction.

Lines 134-135 and 244-245. Please write out what P-I curve stands for. You might need to write out the growth function JPi(PAR,T,N) to make this part understandable.

Lines 141-143. You write that both mortality and metabolic losses become detritus or DON. Is there really no excretion of DIN?

Line 204. You use a spin-up of 30 years. Is this enough to reach steady state? If not, please elaborate on why you chose this time period and how the transient state it might affect the results.

Lines 158-159. Missing explanation for subscript mathrm in PmathrmS and PmathrmL

Lines 258-259. "In deep water (between 100 and 1000 m) most of the plankton compartments (Figure 4) present mild to strong relative responses." I am not sure what you mean here, what could they show other than mild to strong responses? Maybe reformulate to, e.g. "...relative responses vary from mild to strong..."

Line 262 and 268. Please change Appendix "C" to "D"

Line 263. ":" should be "." or "Concentrations" should be "concentrations"

Lines 272-273. Please remove "top" from "top right/left", as there is only one row of figures.

Line 276. Please add "zone" to "coastal upwelling zone"

Lines 284-285, Fig 6. Units seem to be mixed (m-2 and m-3). Please check.

Lines 301-308: Although this paragraph is correct, I had to read it several times before

understanding what you mean. Would there be a simpler way of expressing this?

Section 4.1 (e.g. lines 337-342). How come you use a range of mortalities that is much lower than the one estimated by Hirst and Kiørboe (2002), when the simulated zooplankton concentrations are an order of magnitude higher than measured ones, and the sensitivity analysis shows that increasing mortality decreases biomasses? I see your argument that measured data is uncertain and may be lower that actual mass, but I see no reason to assume that it would be an order of magnitude too low? It would be interesting to see if you could improve model-data fit by using the high mortality rate estimated in the field (0.19/day, Hirst and Kiørboe, 2002) or used in other models (0.25 mmol N m-3/day, Lima and Doney, 2004).

Line 408. It would help the reader if you would add a short description of the Getzlaff and Oschlies (2017) study the first time it is mentioned in the discussion, e.g. what system(s) did they study, what kind of model (global/regional), same methods of sensitivity analysis? I see now that this comes a few paragraphs down. Maybe you could move part of it up to line 408?

Line 464. Please change "affects" to "affect".

Reference list. References are sufficient and relevant.

Line 631. The doi for Jose et al. 2017 leads to another article.

Appendices. Contents of Appendices are relevant for the study. Please place text and figures of each Appendix in a logical order.

---

## Referee Comment (RC3) · Anonymous Referee #3 · 22 Jan 2021

General Comments

The manuscript describes the responses of the plankton ecosystem in the Eastern Tropical South Pacific to different scenarios of small pelagic fish abundance using a coupled physical-biogeochemical model in a regional configuration. Changes in fish predation are simulated by changing mortality rate of zooplankton compartments in the biogeochemical model. This simple method provides an insight of the ecosystem's response to fluctuations in small pelagic fish biomass.

The manuscript addresses relevant scientific questions within the scope of the journal.

The study is based on a previous work (Getzlaff and Oschlies, 2017) carried out on a global scale, using a different model.

The overall presentation is well structured and clear.

The results are discussed in an appropriate and balanced way.

Substantial conclusions are reached but need to be confirmed with an end-to-end model.

The title clearly reflects the contents of the paper.

The abstract provide a concise and complete summary.

The amount and quality of supplementary material is appropriate.

This study has 2 weaknesses:

1/ the evaluation of the plankton compartments is poor. There is little data and the comparison is not convincing. However, the difficulty of comparing model and observations is well discussed. Are there no more in the area?

2/ this study would have deserved some prior improvements: DVM implementation and a tuning of the model. However these two points are mentioned as weaknesses in the discussion.

Specific comments

The reference study of Getzlaff and Oschlies (2017) is based on a simulation that has been running for 300 years. It shows that the Tropics are really long to reach a balance and that the difference between an experiment (high, low scenario) and the reference can changes sign between the first decades of simulation and the rest (Getzlaff and Oschlies, 2017; see Figures 2, 3). So what is the strategy justifying a 30 year climatological simulation ? What are the reasons for this choice? Does the model

reach a state of equilibrium ? Please provide a figure with the time evolution of the main biomasses and fluxes, as in Getzlaff and Oschlies (2017)

I wonder about the relevance of these results in an end-to-end ecosystem. In the high scenario, the flow of energy and matter return immediately to the detritus pool and feed the microbial loop instead of being transferred to higher trophic levels and take longer time to return to the microbial loop. Won't this difference affect the conclusions of this study? This point deserves to be discussed.

A paragraph is missing in the Introduction to describe plankton groups in the study area : the spatial distribution, the succession from the coast to the open ocean... It is disseminated throughout the paper, but it would be clearer to have it in the Introduction.

This manuscript is based on the Getzlaff and Oschlies (2017) study. This latter should be described in Section 2.4 or the first time you discuss it in the discussion section. I mean: specify the area of the study, a different model, the method, the scenario, a 300 year simulation. We learn the main elements of this study but too late in the text.

L42: please indicate that this calculation is valid excluding any non-linearity.

Section 2.3: please modify the title to "Zooplankton comparison" or "zooplankton evaluation", because we can't say it is a validation.

Section 2.3: The model is compared to data between February 10 to March 3, 2013. Which model data are used for the comparison ? an annual mean for the last year of simulation ? a monthly average ? a daily average ? Please specify.

Figure 2: the comparison is not really convincing. Why not show the comparison in log transform as in Appendix C ? This would be justified, as biomasses often have a log distribution.

L199: what is the width of this box? Because Figure 2 shows that the zoo maximum is not to the coast but offshore ($\sim$ 50 km offshore). Is this maximum included in the box ?

Figure 3: please add "in the reference scenario" at the end of the legend.

L209: "coastal upwelling" section: do you mean the white line or the coastal blue box in Figure 1 ? if it is the latter, please change to "coastal upwelling region".

L244-250: Is the spatial pattern of modelled plankton realistic? Is this distribution found in observations, described in literature? Is plankton succession from the coast towards the open ocean typical of EBUS ?

L255: why are deep large detritus increased in the A_low scenario ?

Section 3 and Figure 4: I wonder what the description of the deep zone (100-1000m) provides because the analysis focuses on the surface layer. I would remove that part. This would simplify Figure 4 and remove the questions about the strong differences in phyto and zoo found at depth, even if this is explained in the text. I think it would simplify the message.

Figure 5: please specify that 12°S section refers to the white line in Figure 1.

L339-340: The mortality rates estimated for linear assumption are lower than the 0.19 d-1 estimated by Kiørboe (2002) at 25°C, but there are close to the 0.062 d-1 estimated at 5°C in the same study. Why compare to the first estimate and not the second? What is the temperature in the region ?

L340: "This indicates that the model may not include all potential sources of variability." This sentence should be changed. Of course the model does not take into account all kinds of variability, this must be mentioned, but variability difference between the model and the data cannot be summarized by this sentence. Several other reasons should be mentioned: 1/ In-situ observations represent a snapshot of the ocean while the model outputs are an average (daily, ... not specified in the text). 2/ there is a crucial lack of data to make a robust assessment. 3/ The sampling methods do not allow for a representative sampling. 4/ The scenario uses a climatological simulation, without taking into account inter-annual variability.

L407-408: Are these numbers for the coastal upwelling area or the full domain ?

L408-409: The sentence should be rephrased. It suggests that there are several compartments of zooplankton in their study, whereas there is only one, so obviously "the response depends only on one zooplankton size class". In fact, since this manuscript is based on the Getzlaff and Oschlies (2017) study, this latter should be described before being discussed.

L409-412: are these numbers for the global ocean or for your specific area ?

L407-412: model with 1 plankton compartment = mild changes. Model with 2 compartments = more pronounced changes. What would be expected with 3 plankton compartments ? Can we think that the more plankton compartments there are in the biogeochemical model, the greater the change in plankton biomass ?

L428-429: I do not understand this sentence. Figure 5 shows a maximum at the coast and not at the transition from coast to open ocean.

Table 1: explain in the legend why "Global" and "Full" are put together, same for "Tropics" and "Oligotrophic", "Southern ocean" and "Coastal Upwelling"

L445-447: I am sorry, I do not find the same numbers. Could you detail them please ?

L 494: I do not understand. Figure 4 shows that grazing on Zs is not affected

L 497: ENSO seems to be the main factor but is not discussed.

Technical corrections

L41: the units "Mt" has not been defined above, please define it or use the full name.

L119: correct "dissolved"

L129-130: correct the exudation symbol

L272: please remove "top" when you refer to Figure 5

L273: please remove "top" when you refer to Figure 5

Figure 6: please check units

L337: please change "ZL" to "ZL"

L339: "estimated "instead of "estimate"

Appendices: Please place text together with figures for each Appendix.

---

## Author Comment (AC1) · 22 Feb 2021

**Comment**:

Dear authors,

General comments

The manuscript describes the responses of the ETSP region in terms of model experiments by changing mortality rates of the two zooplankton compartments that the model

includes. It is an interesting study and it is related to the scope of the journal.

Overall, I find the model does not match very well the observations at surface level (approx. 20m depth. This is indeed discussed in the manuscript and special attention should be put in these differences, more details should be provided about the sources for these differences. The main strength of this manuscript is represented by the analysis of the experimental design and the responses of the different compartments in the model and how they relate to the region chosen in the domain. I think this manuscript is within the standards of excellence of this journal in terms of scientific quality and significance.

**Response**:

Dear referee,

The authors thank you for your useful comments and support for this paper. We will put more emphasis on the sources of differences between the model and observations in Sect. 4.1.

We note that in the analysis of scenarios A, which serve as complement for experiments B, there was a mistake in the weighting of the time steps when calculating the annual average of the concentrations. This has now been corrected and affects slightly Fig. 4 and Fig. 10 of the paper (see Figures 1 and 2 in this response). For Fig. 4 in the paper we now only present the surface concentrations of organic compartments, to follow the suggestion by referee three (presented here as Figure 3). These changes do not change any of our conclusions.

**Comment**:

Abstract

My main comment from the abstract and throughout the manuscript is to be consistent with the names you use for the compartments of the model: either use 'large' and 'small' zooplankton or use 'meso' and 'micro' zooplankton. The same comment goes

for the compartments of phytoplankton: nano- and micro-phytoplankton or large and small phytoplankton. If they do not mean the same, you should state this, but if they do, you must be consistent with the use of those names.

**Response**:

We will standardise the language and refer to modelled plankton only as "large" and "small" while keeping the term "mesozooplankton" when referring to observations. This will be explained in the introduction of the revised manuscript.

**Comment**:

Introduction

L29 'presumably due to sensitivity to environmental variability' – can you specify what environmental variability you refer to?

L32-38 This paragraph should be rephrased or use a connector when you explain the two species.

L50-51 The last sentence in the paragraph should be rephrased for better clarity as it is difficult to understand.

L58 Can you specify what is the definition of linear and quadratic mortality?

**Response**:

In line 29 we mean the cessation of upwelling caused by El Niño, and will rephrase the sentence accordingly. We will also rephrase lines 32-38 and 50-51, and explain linear and mortalities including their mathematical expressions in the revised manuscript.

**Comment**:

Methods

Figure 1 – in the introduction you mention that anchovy spans from northern Peru to Talcahuano. Can you add landmarks in the map to show where is Peru and Chile? And

why did you choose that domain for each region (O, C, F)?

**Response**:

Politic divisions will be added on Fig. 1 of the paper (see Figure 4).

The full domain was chosen to have an overview on the whole model response. C and O where picked to examine differences in a region with high and low nutrients. The coastal upwelling region was chosen in an area of high nutrient concentrations. Since the upwelling system of Peru is quite heterogenous with lots of mesoscale processes, we restricted our high nutrient region to the very coastal upwelling area where the concentration of large phytoplankton is high. The oligotrophic region was picked as far as possible from the nutrient rich areas along the Equator and along the coast, but apart from the domain boundary to avoid boundary effects. We will extend the description of these regions, and extended the explanation why we picked them at the end of section 2.5.

**Comment**:

L108 – 126 When you define the compartments for phytoplankton and zooplankton, it would be good to give a quantitative definition for each compartment. For example, in terms of size class: does microzooplankton include all species that are <200 $\mu$m? And the same for the other compartments.

**Response**:

The model does not have a size parameter in the strict sense (as, for example, size-dependent allometric rates) but instead it tries to replicate groups that occupy niches of the large and small communities by simulating the nitrogen fluxes going in and out of each compartment. Therefore, the small zooplankton compartment represents the whole zooplankton community smaller than about 200 $\mu$m and large zooplankton the community larger than that. Similarly, large and small phytoplankton aim at representing communities larger and smaller than about 20 $\mu$m. We will clarify this in Sect.

[Figure]

2.2.

**Comment**:

L139 – 146 Do you consider temperature dependence in zooplankton processes? How can this affect your results if you would include it? Or justify/discuss why not having it and the effect this could have in your results if it is relevant.

**Response**:

The BioEBUS model in its standard configuration does not include any temperature-dependent zooplankton rates. We expect, that if temperature-dependent grazing of zooplankton was implemented, the loophole for phytoplankton growth in the cold waters of the coastal upwelling region (on which we comment in Sects. 3.1 and 4.2) would be even widened, amplifying the "spatial succession" we observed. However, temperature might also affect zooplankton metabolism, with colder temperatures decreasing, for example, its excretion rates, which could mute these effects again. We will comment on this in Sect. 4.2.

**Comment**:

L205 It might be worth briefly describing the way the model calculates the nitrogen and state variables fluxes or at least reference a paper that describes those processes.

**Response**:

The model considers several nitrogen processes including anammox, denitrification and nitrification. Because we did not focus on these processes in the study, we kept their explanation to the minimum. The processes are described in detail in the paper by Gutknecht et al. (2013) which is based on the formulation by Yakushev et al. (2007). We will expand in the revised version of the paper.

**Comment**:

L258-259 P_S shows an important difference in deep water for the A_high and B_low

experiments, are those differences negligible? It might be good to at least mention or have a brief explanation about those differences.

**Response**:

The absolute differences are negligible (0.00004 and 0.00001 in O, 0.00004 and -0.00004 in F, and -0.00019 and -0.00034 mmol N m$^{-2}$ for A_high and B_low respectively). Based on the suggestions made by reviewer 3 we now decided to remove the deep layer (100 to 1000 m) from Fig. 4 of the paper (see Fig. 3) because this is not an important component of the paper and complicates the figure.

**Comment**:

L321 Please give some examples about the non-linear processes you refer to in this sentence.

**Response**:

For example, a quadratic zooplankton mortality exacerbates the reduction in zooplankton biomass when concentrations are very high, and prevents its extinction at very low concentrations. In addition, the multiple resources form of the Holling type II grazing function allows the predator to modify its grazing preference towards the most abundant prey (Fasham et al., 1999). Finally, Lima et al. (2002) noted that coupled physical and food web models can transition from equilibrium to chaotic states under even small changes in their parameters. We will mention this in Sect. 4.1.

**Comment**:

L333-334 Consider not only sampling issues in the differences in Figure 2. What about assumptions in the model? Or bad parameterisation (not only in terms of zooplankton mortality)? Also, you should mention if using a quadratic mortality rate instead of a linear one could produce differences in your results.

**Response**:

For this study we changed the quadratic mortality term of our model. Systems with a non-density dependant, or línear, mortality rate, respond to perturbations in a "reactive" way, as defined by Neubert et al. (2004), drifting away from equilibrium. We might have expected a stronger impact if we had manipulated the linear mortality due to the lack of the buffer by the density dependency of the quadratic term. We will mention this in section 4.1.

We recall that the observations are likely biased low since they do not include the whole taxonomic range, but mainly crustaceans. However, indeed it is very likely that the model could be improved by a tuning or calibration exercise that targets at a good match between observed and simulated zooplankton concentrations. Despite the complexity of the model, the considerable uncertainty of model parameters, and the sparsity of observations that can constrain these parameters, this is a complex task (see, e.g., Kriest et al., 2017). Therefore, we have refrained from this effort for the present, but aim at provide a better calibrated model in the future. We will add some comments on this in the discussion section.

**Comment**:

L340 It is not clear to me if you just assumed the values given by Hirst & Kiorboe (2002) for you experimental design. Could the values given by Hirst and Kiorboe (2002) be affected by temperature? Is 25degC a realistic value for your domains (see Fig.1). How much change could be expected in terms of zooplankton mortality rates for 2degC difference? Or is that negligible ?

**Response**:

We did not assume Hirst and Kiørboe (2002) values on our experimental design, but only use these values for comparison. The values provided by Hirst and Kiørboe (2002) are indeed affected by temperature. At $5°$ C, a mortality of 0.065 d$^{-1}$, which is closer to our reference simulation estimate, was reported on the same paper. We will compare against this value in the discussion section. Our modelled region is indeed slightly

colder than 25° C (see Figure 1). However, the zooplankton metabolic processes, including mortality, are not affected by temperature in our model so a 2° C change would not have any impact on zooplankton mortality.

**Comment**:

Conclusions

L497 You mention some effects from ENSO but there is not much discussion associated to this in section 4.

**Response**:

The ROMS setup presented here is climatological and does not include ENSO and its effects. Because in this study we are mainly interested on the potential effects small pelagic fish may have on the plankton community, rather than the cause for their fluctuations, we will skip the mention of ENSO in a revised version of the paper.

**Comment**:

Technical corrections

L28 The name of that city is Talcahuano, not Telcahuano.

L119 correct 'dissolved'

L130 correct the symbol for the exudation fraction for eqs. 2, 3 as they do not match both equations.

L157 – 158 correct the typo of the terms PmathrmS, PmathrmL.

L167 missing a parenthesis ending after Gorsky et al. 2010.

**Response**:

All technical corrections will be implemented.

**References**

Fasham, M. J. R., Boyd, P. W., and Savidge, G.: Modeling the relative contributions of autotrophs and heterotrophs to carbon flow at a Lagrangian JGOFS station in the Northeast Atlantic: the importance of DOC, Limnology and Oceanography, 44(1), 80–94, https://doi.org/10.4319/lo.1999.44.1.0080, 1999.

Gutknecht, E., Dadou, I., Le Vu, B., Cambon, G., Sudre, J., Garçon, V., Machu, E., Rixen, T., Kock, A., Flohr, A., et al.: Coupled physical/biogeochemical modeling including O2-dependent processes in the Eastern Boundary Upwelling Systems: application in the Benguela, Biogeosciences, 10, 3559–3591, https://doi.org/10.5194/bg-10-3559-2013, 2013a.

Hirst, A. and Kiørboe, T.: Mortality of marine planktonic copepods: global rates and patterns, Marine Ecology Progress Series, 230, 195–209, https://doi.org/10.3354/meps230195, 2002.

Kriest, I., Sauerland, V., Khatiwala, S., Srivastav, A., Oschlies, A.: Calibrating a global three-dimensional biogeochemical ocean model (MOPS-1.0), 10, 127–154, https://doi.org/10.5194/gmd-10-127-2017, Geoscientific Model Development, 2017.

Lima, I. D., Olson, D. B., and Doney, S. C.: Intrinsic dynamics and stability properties of size-structured pelagic ecosystem models, Journal of Plankton Research, 24, 533–556, https://doi.org/10.1093/plankt/24.6.533, 2002.

Neubert, M. G., Klanjscek, T. and Caswell, H.: Reactivity and transient dynamics of predator–prey and food web models, Ecological Modelling, 179(1), 29–38, https://doi.org/10.1016/j.ecolmodel.2004.05.001, 2004.

Yakushev, E., Pollehne, F., Jost, G., Kuznetsov, I., Schneider, B., and Umlauf, L.: Analysis of the water column oxic/anoxic interface in the Black and Baltic seas with a numerical model, Marine Chemistry, 107, 388–410, https://doi.org/10.1016/j.marchem.2007.06.003, 2007.
* * *
[Figure]

**Fig. 1.** Same as Fig. 4 of the paper after correcting the averaging weights in experiments A.

[Figure]

**Fig. 2.** Same as Fig. D1 of the paper after correcting the averaging weights in experiments A.

[Figure]

**Fig. 3.** Same as Fig. 4 of the paper after correcting the averaging weights in experiments A and removing the deep water (100 to 1000 m) layer.

**Fig. 4.** Same as Fig. 1 of the paper including political division.

---

## Author Comment (AC2) · 22 Feb 2021

**Comment**:

GENERAL COMMENTS

This sensitivity study of zooplankton mortality in a regional physical-biogeochemical model of the Eastern Tropical South Pacific provides valuable insights into the responses of ecosystem dynamics and biogeochemistry caused by changes in zooplankton abundance. The relatively simple approach enables disentangling the main drivers

of change and their relative strength, showing e.g. that large zooplankton assert a top-down control on both small zooplankton and phytoplankton, causing asymmetrical ecosystem responses.

The paper is well-written, fluent and concise, clearly outlining the methods and assumptions, except for some minor issues addressed in the specific comments. It builds on the conceptual framework of Gezlaff and Oschlies (2017), but applies it to a new study system on a different spatial scale using a different model. The comparison of outcomes from the current regional study to the previous global study is one of the main strengths of this paper.

The model used has been fully described and partially validated in previous papers, with the most relevant equations and parameters repeated in this paper. However, I would like clarifications to some parts as written in the specific comments.

I also have two concerns regarding the zooplankton model formulations. It seems to exclude two important functions for zooplankton dynamics: i) excretion of DIN (ammonium) and ii) temperature-dependence of zooplankton growth/metabolism (see e.g.Tian 2006, Ecol. Model.; Richardson 2008, ICES J. Mar. Sci.). The first might be a mistake in the text (lines 141-143), as DIN excretion is included in the model as described by Gutknecht et al. (2013). If these functions are indeed omitted, I would like the authors to motivate why and discuss if they think this could have major implications for the simulated plankton biomasses and biogeochemistry of the system.

The validation of zooplankton biomass over depth shows that the model is not a very accurate representation of the specific system, especially since simulated surface concentrations are one order of magnitude higher than measured ones. However, the authors provide a thorough discussion of this, and I think the results are still sufficient to support the interpretations and conclusions drawn from the study. I believe the results from this study are a valuable contribution both to the general scientific understanding of real-world plankton dynamics and their effects on the ecosystem, as well as to the

development of commonly used biogeochemical ecosystem models.

In summary, I think this paper lives up to the standards of Biogeochemistry and recommend that it should be published after minor revisions.

**Response**:

Dear referee,

The authors thank you for your useful comments and support for our paper. Part of the metabolic losses of zooplankton do become DIN, we will correct this. Zooplankton metabolism and mortality are not affected by temperature. We will comment of this in the revised version of the paper.

We note that in the analysis of scenarios A, which serve as complement for experiments B, there was a mistake in the weighting of the time steps when calculating the annual average of the concentrations. This has now been corrected and affects slightly Fig. 4 and Fig. D1 of the paper (see Figures 1 and 2 in this response). For Fig. 4 in the paper we now only present the surface concentrations of organic compartments, to follow the suggestion by referee three (presented here as Figure 3). These changes do not change any of our conclusions.

**Comment**:

SPECIFIC COMMENTS AND TECHNICAL CORRECTIONS

Title clearly reflects the contents of the paper.

Abstract provides a concise and complete summary.

Equations 2, 3 and line 131. Please check the symbol for exudation fraction.

**Response**:

We will fix the symbol in Eq. 3.

**Comment**:

Lines 134-135 and 244-245. Please write out what P-I curve stands for. You might need to write out the growth function JPi(PAR,T,N) to make this part understandable.

**Response**:

We will write down that P-I stands for photosynthesis-irradiance and explain the growth function.

**Comment**:

Lines 141-143. You write that both mortality and metabolic losses become detritus or DON. Is there really no excretion of DIN?

**Response**:

One fraction of the metabolic losses does become ammonium without further reminer-alization steps in between. We will correct this in the manuscript.

**Comment**:

Line 204. You use a spin-up of 30 years. Is this enough to reach steady state? If not please elaborate on why you chose this time period and how the transient state it might affect the results.

**Response**:

We do not dispose of the high resolution time series of the reference run at the moment. Therefore, we have provided a time series of the final 10 years of the spin-up of simulation A_high (Figure 4). While the deep water might not be in steady state after the 30 years spin up, the upper 100 m are already quite stable after 20 years. The averaged relative changes in concentrations for the same day in different years for the averaged full domain are equal or lower than 0.1 % for the four plankton groups. Nevertheless, concentrations vary more in the smaller regions. For the oligotrophic region, small zooplankton exhibits the minimum averaged relative interannual change (-0.007 %), and large zooplankton the maximum (5.2 %); while in the coastal upwelling
region, the minimum and maximum averaged relative interannual changes are 0.3 %
for large zooplankton , and 9.4 % for small zooplankton respectively. In comparison,
Le Quere et al. (2016) reported that the 10 plankton functional groups in their model
(only four in ours), reached equilibrium after only 3 years.

**Comment**:

Lines 158-159. Missing explanation for subscript mathrm in PmathrmS and PmathrmL

**Response**:

The mathrm subscript label was an error in the LaTex code due to the line break. It will
be fixed to $P_S$ and $P_L$.

**Comment**:

Lines 258-259. "In deep water (between 100 and 1000 m) most of the plankton com-
partments (Figure 4) present mild to strong relative responses." I am not sure what
you mean here, what could they show other than mild to strong responses? Maybe
reformulate to, e.g. "...relative responses vary from mild to strong: : :"

**Response**:

We will remove the explanation on plankton groups response in the deep water, and
remove this layer from Fig. 4 of the paper as suggested by another referee, because
this does not provide anything to the conclusions of the study and the deep layers make
Fig. 4 unnecessarily complicated.

**Comment**:

Line 262 and 268. Please change Appendix "C" to "D"

Line 263. ":" should be "." or "Concentrations" should be "concentrations"

Lines 272-273. Please remove "top" from "top right/left", as there is only one row of
figures.

**Response**:

We will make these corrections.

**Comment**:

Line 276. Please add "zone" to "coastal upwelling zone"

**Response**:

We will add "region" here because this is how we refer to C in the whole paper. There was a mistake in Sect. 2.5 where we called it "coastal upwelling section". This will also be corrected.

**Comment**:

Lines 284-285, Fig 6. Units seem to be mixed (m-2 and m-3). Please check.

**Response**:

We will correct the units referring to Fig. 6 in the text. The correct units are mmol N m$^{-2}$ d$^{-1}$ and mmol N m$^{-2}$

**Comment**:

Lines 301-308: Although this paragraph is correct, I had to read it several times before understanding what you mean. Would there be a simpler way of expressing this?

**Response**:

We will rephrase this paragraph and also refer to the nitrogen loss due to zooplankton mortality as $\mu_{Z_i} \cdot [Z_i]^2$ to simplify the text.

**Comment**:

Section 4.1 (e.g. lines 337-342). How come you use a range of mortalities that is much lower than the one estimated by Hirst and Kiørboe (2002), when the simulated zooplankton concentrations are an order of magnitude higher than measured ones,

and the sensitivity analysis shows that increasing mortality decreases biomasses? I see your argument that measured data is uncertain and may be lower that actual mass, but I see no reason to assume that it would be an order of magnitude too low? It would be interesting to see if you could improve model-data fit by using the high mortality rate estimated in the field (0.19/day, Hirst and Kiørboe, 2002) or used in other models (0.25 mmol N m-3/day, Lima and Doney, 2004).

**Response**:

The zooplankton mortality has a rather broad range of values in biogeochemical models, evidencing the little information available to constrain it. The value by Lima and Doney (2004) lies in the upper range of mortality values found in the literature. The estimate by Hirst and Kiørboe (2002) at 25° C is about twice as high as our estimate for the high mortality scenario. On the other hand, the estimate at 5° C lies between our low mortality and reference scenario estimate.

The current set of parameters of the model were adjusted mainly to represent nutrient and oxygen concentrations in this region, and the model at that stage was not tuned to match plankton well (see Jose et al., 2017). Indeed, this is the first time that we evaluate any of the zooplankton compartments for the Easter Tropical South Pacific region. In addition, the $\pm 50$ % change in mortality, aside the comparison to observations, was also picked to compare against the study by Getzlaff and Oschlies (2017). The results of this study have indeed drawn a direction for the further improvement of the model. However, a stronger change in zooplankton mortality than the values used here may require further adjustment of other parameters. In such a complex model with considerable uncertainty in its parameters and little observations to compare with, a proper tuning is a complex task (see, e.g., Kriest et al., 2017). Therefore, we have not included it for the present study but aim at improving the model for further work. Nevertheless in the revised version of the paper we will expand the comparison to also mention the mortality estimates by Hirst and Kiørboe (2002) at 5° C and will discuss further avenues of model improvement with regard to zooplankton in the revised manuscript.

[Figure]

**Comment**:

Line 408. It would help the reader if you would add a short description of the Getzlaff and Oschlies (2017) study the first time it is mentioned in the discussion, e.g. what system(s) did they study, what kind of model (global/regional), same methods of sensitivity analysis? I see now that this comes a few paragraphs down. Maybe you could move part of it up to line 408?

**Response**:

We will include a brief description of the study by Getzlaff and Oschlies (2017) in line 407 and rephrase the following lines.

**Comment**:

Line 464. Please change "affects" to "affect".

**Response**:

We will correct this.

**Comment**:

Reference list. References are sufficient and relevant.

Line 631. The doi for Jose et al. 2017 leads to another article.

**Response**:

We will change to the correct DOI: 10.5194/bg-14-1349-2017

**Comment**:

Appendices. Contents of Appendices are relevant for the study. Please place text and figures of each Appendix in a logical order.

**Response**:

We will try to position every figure next to the text of its corresponding appendix section. We will exchange the positions of Appendix A and B to their order of appearance in the text.

**References**

Getzlaff, J. and Oschlies, A.: Pilot Study on Potential Impacts of Fisheries-Induced Changes in Zooplankton Mortality on Marine Biogeochemistry, Global Biogeochemical Cycles, 31, 1656–1673, https://doi.org/10.1002/2017GB005721, 2017.

Hirst, A. and Kiørboe, T.: Mortality of marine planktonic copepods: global rates and patterns, Marine Ecology Progress Series, 230, 195–209, https://doi.org/10.3354/meps230195, 2002.

Jose, Y. S., Dietze, H., and Oschlies, A.: Linking diverse nutrient patterns to different water masses within anticyclonic eddies in the upwelling system off Peru, Biogeosciences, 14, 1349–1364, https://doi.org/10.5194/bg-14-1349-2017, 2017.

Kriest, I., Sauerland, V., Khatiwala, S., Srivastav, A., Oschlies, A.: Calibrating a global three-dimensional biogeochemical ocean model (MOPS-1.0), 10, 127–154, https://doi.org/10.5194/gmd-10-127-2017, Geoscientific Model Development, 2017.

Le Quere, C., Buitenhuis, E. T., Moriarty, R., Alvain, S., Aumont, O., Bopp, L., Chollet, S., Enright, C., Franklin, D. J., Geider, R. J., Harrison, S. P., Hirst, A. G., Larsen, S., Legendre, L., Platt, T., Prentice, I. C., Rivkin, R. B., Sailley, S., Sathyendranath, S., Stephens, N., Vogt, M., Vallina, S. M.: Role of zooplankton dynamics for Southern Ocean phytoplankton biomass and global biogeochemical cycles, 14, 4111–4133, https://doi.org/10.5194/bg-13-4111-2016, Biogeosciences, 2016.

Lima, I. D. and Doney, S. C.: A three-dimensional, multinutrient, and size-structured ecosystem model for the North Atlantic, Global Biogeochemical Cycles, 18, https://doi.org/10.1029/2003GB002146, 2004.

[Figure]

**Fig. 1.** Same as Fig. 4 of the paper after correcting the averaging weights in experiments A.

[Figure]

**Fig. 2.** Same as Fig. D1 of the paper after correcting the averaging weights in experiments A.

**Fig. 3.** Same as Fig. 4 of the paper after correcting the averaging weights in experiments A and removing the deep water (100 to 1000 m) layer.

[Figure]

[Figure]

**Fig. 4.** Time series from year 21 to year 30 of integrated concentrations (upper 100 m) averaged over space (thick lines), and percentage difference between every point in time and the same date one year later

---

## Author Comment (AC3) · 22 Feb 2021

**Comment**:

General Comments

The manuscript describes the responses of the plankton ecosystem in the Eastern Tropical South Pacific to different scenarios of small pelagic fish abundance using a coupled physical-biogeochemical model in a regional configuration. Changes in fish predation are simulated by changing mortality rate of zooplankton compartments in

the biogeochemical model. This simple method provides an insight of the ecosystem's response to fluctuations in small pelagic fish biomass.

The manuscript addresses relevant scientific questions within the scope of the journal. The study is based on a previous work (Getzlaff and Oschlies, 2017) carried out on a global scale, using a different model.

The overall presentation is well structured and clear.

The results are discussed in an appropriate and balanced way.

Substantial conclusions are reached but need to be confirmed with an end-to-end model.

The title clearly reflects the contents of the paper.

The abstract provide a concise and complete summary.

The amount and quality of supplementary material is appropriate.

This study has 2 weaknesses:

1/ the evaluation of the plankton compartments is poor. There is little data and the comparison is not convincing. However, the difficulty of comparing model and observations is well discussed. Are there no more in the area?

2/ this study would have deserved some prior improvements: DVM implementation and a tuning of the model. However these two points are mentioned as weaknesses in the discussion.

**Response**:

Dear referee,

The authors thank you for your valuable comments to improve the analysis and quality of the paper.

We agree of the importance of including more data in the comparison. Fig. 5 shows a comparison between the model (right) and the Eastern Tropical South Pacific region of the global mesozooplankton dataset by Moriarty and O'Brien (2013); O'Brien and Moriarty (2012) (left). We transformed the observation values provided in the dataset to nitrogen units assuming a carbon to nitrogen ratio by weight of 4.9 (Kiørboe, 2013) and a nitrogen molar mass of 14 g/mol. Both model and observations are averaged over the whole year and over the upper 100 m depth. Model values are generally higher than observations. However, please note that the observations are sparse and in many cases there is only one data point available for the whole water column. Therefore, the averages may not be representative of the whole water column. We can include this figure in the appendix of the paper if recommended. Because the available data is from heterogenous sources, and often on a coarse spatial grid, it requires a very careful analysis for comparison with our finely resolved model. A more careful and in depth validation, including tuning of the model will be presented elsewhere.

We note that in the analysis of scenarios A, which serve as complement for experiments B, there was a mistake in the weighting of the time steps when calculating the annual average of the concentrations. This has now been corrected and affects slightly Fig. 4 and Fig. D1 of the paper (see Figures 1 and 2 in this response). For Fig. 4 in the paper we now only present the surface concentrations of organic compartments, to follow your suggestion further down (presented here as Figure 3). These changes do not change any of our conclusions.

**Comment**:

Specific comments

The reference study of Getzlaff and Oschlies (2017) is based on a simulation that has been running for 300 years. It shows that the Tropics are really long to reach a balance and that the difference between an experiment (high, low scenario) and the reference can changes sign between the first decades of simulation and the rest

(Getzlaff and Oschlies, 2017; see Figures 2, 3). So what is the strategy justifying a 30 year climatological simulation ? What are the reasons for this choice? Does the model reach a state of equilibrium ? Please provide a figure with the time evolution of the main biomasses and fluxes, as in Getzlaff and Oschlies (2017).

**Response**:

By using a higher resolution model, we can compare the effect of the same strategy at a regional scale with higher physical complexity that is not addressed by the global model. The compromise is that the high resolution model is more computationally demanding and it is technically impossible to run it for 300 years.

Currently the high temporal resolution simulation results are not available for all experiments. Therefore, we cannot provide a figure as in Getzlaff and Oschlies (2017) study. However, we have provided the averaged concentrations over the final 10 years of the simulation, as well as the interannual changes (percentage change between every point in time and one year later, Figure 4). As it can be seen in the figure, there is no major trend in increase or decrease of biomass for any of the plankton groups.

Furthermore, the 300 years time series in Getzlaff and Oschlies (2017) model starts after a spin up time of 10 000 years with the parameters of the reference scenario. Therefore, the 300 years run in their model reflects the re-stabilization of the model from the reference scenario conditions to the ecosystem conditions with a shifted zooplankton mortality. On the other hand, we spun up the model with the already changed mortality.

**Comment**:

I wonder about the relevance of these results in an end-to-end ecosystem. In the high scenario, the flow of energy and matter return immediately to the detritus pool and feed the microbial loop instead of being transferred to higher trophic levels and take longer time to return to the microbial loop. Won't this difference affect the conclusions of this

study? This point deserves to be discussed.

**Response**:

In an equilibrium state this would not be a problem since there is a constant turnover of nutrients. In a non-equilibrium state, the time that nutrients spend as part of larger animals biomass would further the gap between nutrients consumption by phytoplankton and their replenishment affecting phytoplankton growth rate and potentially the blooming timing. However, this should not be a problem in the coastal upwelling region because nutrients are highly concentrated here. This is not the case for the oligotrophic region although in our study, this region presents the weakest response. Furthermore, small pelagic fish concentrate mainly in the highly productive upwelling region rather than in the oligotrophic waters offshore. On the other hand, fish and larger animals are highly mobile and are not constantly drifted by advection as nutrients and plankton do. Therefore, migrations transport nutrients and organic matter in and out of the region in a horizontal (McInturf et al., 2019; Varpe and Fiksen, 2005; Williams et al., 2018) and vertical fashion (Davison et al., 2013; Lavery et al., 2010). We will point this out in Sect. 4.3.

Furthermore, the inclusion of more higher trophic levels in an ecosystem implies an inherent energy loss per additional trophic step (see Ryther, 1969), hence reducing the total trophic transfer efficiency of the ecosystem. On the other hand, large organisms tend to live longer and they are considered to have a more efficient metabolism (Brown et al., 2004). As a consequence, they store in their bodies more biomass relative to their metabolic losses than smaller organisms over their lifespan. Therefore, the turnover rate of organic matter may ultimately also depend on the number of trophic levels above large zooplankton as well as their size and longevity. These are open questions that might be explored with finely resolved ecosystem models. An end-to-end model for the Northern Humboldt Current System is currently work in progress which may help to elucidate some of these questions. Finally, we have seen in this study with only two size classes of zooplankton that the relative abundances of both

zooplankton and phytoplankton changed in relationship to each other when zooplankton mortality changed. Therefore, in a real ecosystem we might expect shifts in the zooplankton size spectrum with the corresponding changes in the trophic transfer efficiency of the ecosystem. A biogeochemical model with a higher resolution in the plankton groups might then be necessary to further explore such changes.

**Comment**:

A paragraph is missing in the Introduction to describe plankton groups in the study area : the spatial distribution, the succession from the coast to the open ocean... It is disseminated throughout the paper, but it would be clearer to have it in the Introduction.

**Response**:

We will include such description in the introduction.

**Comment**:

This manuscript is based on the Getzlaff and Oschlies (2017) study. This latter should be described in Section 2.4 or the first time you discuss it in the discussion section. I mean: specify the area of the study, a different model, the method, the scenario, a 300 year simulation. We learn the main elements of this study but too late in the text.

**Response**:

We will add a description of the methods in Getzlaff and Oschlies (2017) study after line 407.

**Comment**:

L42: please indicate that this calculation is valid excluding any non-linearity.

**Response**:

We will take care of this.

**Comment**:

Section 2.3: please modify the title to "Zooplankton comparison" or "zooplankton evaluation", because we can't say it is a validation.

**Response**:

We will change this to "zooplankton evaluation".

**Comment**:

Section 2.3: The model is compared to data between February 10 to March 3, 2013. Which model data are used for the comparison ? an annual mean for the last year of simulation ? a monthly average ? a daily average ? Please specify.

**Response**:

We will specify in Sect. 2.3 and in the figure caption that the model data is the average from January to March.

**Comment**:

Figure 2: the comparison is not really convincing. Why not show the comparison in log transform as in Appendix C ? This would be justified, as biomasses often have a log distribution.

**Response**:

We wanted to be as fair as possible and not to hide the fact that the model overestimates zooplankton at the surface. However, we do acknowledge that a logarithmic comparison provides valuable information, especially for the deep water where concentrations are low, and this is why we included it in the Appendix.

**Comment**:

L199: what is the width of this box? Because Figure 2 shows that the zoo maximum is not to the coast but offshore ($\sim$ 50 km offshore). Is this maximum included in the box ?

**Response**:

The width of the box is about 40 to 50 Km from the coast. We will mention this in the figure caption and in Sect. 2.5. The large phytoplankton peak is included in this region but not the large zooplankton peak. We choose to do it this way because of the high heterogeneity of the upwelling zone and subsequent transition zone were water is transported offshore. We evaluate the development of such spatial succession in Fig. 5 of the paper.

**Comment**:

Figure 3: please add "in the reference scenario" at the end of the legend.

**Response**:

We will make the addition.

**Comment**:

L209: "coastal upwelling" section: do you mean the white line or the coastal blue box in Figure 1 ? if it is the latter, please change to "coastal upwelling region".

**Response**:

We will change this to "coastal upwelling region" since it refers to the blue box.

**Comment**:

L244-250: Is the spatial pattern of modelled plankton realistic? Is this distribution found in observations, described in literature? Is plankton succession from the coast towards the open ocean typical of EBUS ?

**Response**:

Hutchings (1992) explains spatial successions as a common phenomena in EBUS. We will mention this in the revised manuscript.

**Comment**:

L255: why are deep large detritus increased in the A_low scenario ?

**Response**:

This is most likely an artifact from the averaging error described at the beginning of the response. The increase is not present any more in the corrected figure (Figure 1).

**Comment**:

Section 3 and Figure 4: I wonder what the description of the deep zone (100-1000m) provides because the analysis focuses on the surface layer. I would remove that part. This would simplify Figure 4 and remove the questions about the strong differences in phyto and zoo found at depth, even if this is explained in the text. I think it would simplify the message.

**Response**:

We intended to provide an overview of the whole model response at the beginning of our results section and that is why Fig. 4 also shows the deep water averages. However, we agree that for the purposes of this paper the deep water boxes are not necessary and they distract the reader. Therefore, we will remove them from Fig. 4 in the paper/Fig. 2 here; and will replace it with Fig. 3 of this response.

**Comment**:

Figure 5: please specify that 12° S section refers to the white line in Figure 1.

**Response**:

We will indicate this in the legend of Fig. 5 in the paper.

**Comment**:

L339-340: The mortality rates estimated for linear assumption are lower than the 0.19 d-1 estimated by Kiørboe (2002) at 25° C, but there are close to the 0.062 d-1 estimated at 5° C in the same study. Why compare to the first estimate and not the second? What

is the temperature in the region ?

**Response**:

The temperature in the region lies between 16 and 18° C. We will include also the estimate at 5° C in our comparison.

**Comment**:

L340: "This indicates that the model may not include all potential sources of variability." This sentence should be changed. Of course the model does not take into account all kinds of variability, this must be mentioned, but variability difference between the model and the data cannot be summarized by this sentence. Several other reasons should be mentioned: 1/ In-situ observations represent a snapshot of the ocean while the model outputs are an average (daily, ... not specified in the text). 2/ there is a crucial lack of data to make a robust assessment. 3/ The sampling methods do not allow for a representative sampling. 4/ The scenario uses a climatological simulation, without taking into account inter-annual variability

**Response**:

We will change line 348 to "The observations, on the other hand, are susceptible to sampling errors such as net avoidance, and do not cover the whole taxonomic and size spectrum of mesozooplankton.", and line 351 to "Several sources of variability are not accounted for in the model as it only simulates the most relevant processes in the system. We employ a climatological model which aims at simulating an average state over several years, dismissing interannual variability. Furthermore, we here compare a three months average from January to March, while observations provide only a snapshot of a highly dynamical system."

**Comment**:

L407-408: Are these numbers for the coastal upwelling area or the full domain ?

**Response**:

For the full domain. We will specify accordingly.

**Comment**:

L408-409: The sentence should be rephrased. It suggests that there are several compartments of zooplankton in their study, whereas there is only one, so obviously "the response depends only on one zooplankton size class". In fact, since this manuscript is based on the Getzlaff and Oschlies (2017) study, this latter should be described before being discussed.

**Response**:

We will rephrase that section and include a description of the study by Getzlaff and Oschlies (2017) after line 407.

**Comment**:

L409-412: are these numbers for the global ocean or for your specific area ?

**Response**:

For the global ocean. We will specify accordingly.

**Comment**:

L407-412: model with 1 plankton compartment = mild changes. Model with 2 compartments = more pronounced changes. What would be expected with 3 plankton compartments ? Can we think that the more plankton compartments there are in the biogeochemical model, the greater the change in plankton biomass ?

**Response**:

We would not draw this conclusion from the experiments because there are more differences between the two models. It would be necessary to compare exactly the same model changing only the number of zooplankton compartments. Furthermore, large

zooplankton acts in our model as the main driver of the process, masking the impact of changing the mortality of small zooplankton. In addition, the grazing pressure by small zooplankton has a zig-zag effect on the trophic chain, resulting in an inverse response between the two size classes of zooplankton and phytoplankton.Therefore, we might expect similar responses, if not weaker, in a model with three zooplankton compartments, depending on the ecological role of the third compartment.

**Comment**:

L428-429: I do not understand this sentence. Figure 5 shows a maximum at the coast and not at the transition from coast to open ocean.

**Response**:

In this sentence we do not mean that the bloom occurs in the transition between coastal and open ocean. Instead we mean that the bloom develops at the same time that water is being advected offshore. We will rephrased to: "The phytoplankton bloom, which develops closest to the coast and then is offset while water is transported offshore, can be explained by an imbalance between sources and sinks, triggered by changing environmental conditions."

**Comment**:

Table 1: explain in the legend why "Global" and "Full" are put together, same for "Tropics" and "Oligotrophic", "Southern ocean" and "Coastal Upwelling"

**Response**:

We will add to the legend: "We grouped together "Global" and "Full" because both refer to the whole model domain in each of the two studies. Similarly, "Oligotrophic" and "Tropics" refer to regions characterised by low nutrient concentrations, and "Coastal Upwelling" and "Southern Ocean" are both regions with high nutrient concentrations."

**Comment**:

L445-447: I am sorry, I do not find the same numbers. Could you detail them please ?

**Response**:

We used the formula: $\frac{\mu_i \cdot (Z_i \cdot F)^2 - \mu_R \cdot (Z_R \cdot F)^2}{\mu_R \cdot (Z_R \cdot F)^2} \cdot 100$, where $Z_i$ is the amount of nitrogen in each experiment, $Z_R$ is the nitrogen in the reference scenario, $\mu_i$ and $\mu_R$ are the mortality rates and $F$ is a conversion factor to transform from Tg of nitrogen to mmol N m$^{-3}$. Since $F$ is the same in all cases, this simplifies to: $\frac{\mu_i \cdot Z_i^2 - \mu_R \cdot Z_R^2}{\mu_R \cdot Z_R^2} \cdot 100$.

**Comment**:

L 494: I do not understand. Figure 4 shows that grazing on Zs is not affected

**Response**:

We do not show any grazing fluxes in Fig. 4 of the paper. This sentence refers to Fig. 6. We will add a reference to Section 3.3 here.

**Comment**:

L 497: ENSO seems to be the main factor but is not discussed.

**Response**:

The investigation of ENSO effects are not the main purpose of the paper. We mentioned it as a possible cause of fluctuations in small pelagic fish. However, this paper does not address the causes of such fluctuations, but rather focuses on the consequences. Therefore, we will remove the sentence mentioning ENSO in the conclusions section to avoid confusions.

**Comment**:

Technical corrections

L41: the units "Mt" has not been defined above, please define it or use the full name.

L119: correct "dissolved"

L129-130: correct the exudation symbol

L272: please remove "top" when you refer to Figure 5

L273: please remove "top" when you refer to Figure 5

Figure 6: please check units

L337: please change "ZL" to "ZL"

L339: "estimated "instead of "estimate"

Appendices: Please place text together with figures for each Appendix.

**Response**:

We will implement all technical corrections.

**References**

Allgeier, J. E., Burkepile, D. E., and Layman, C. A.: Animal pee in the sea: consumer-mediated nutrient dynamics in the world's changing oceans, Global Change Biology, 23(6), 2166–2178, https://doi.org/10.1111/gcb.13625, 2017.

Brown, J. H., Gillooly, J. F., Allen, A. P., Savage, V. M., and West, G. B.: Towards a metabolic theory of ecology, Ecology, 85(7), 1771–1789, https://doi.org/10.1890/03-9000, 2004.

Davison, P. C., Checkley, D. M., Koslow, J. A., and Barlow, J.: Carbon export mediated by mesopelagic fishes in the northeast Pacific Ocean, Progress in Oceanography, 116, 14–30, https://doi.org/doi, 2013.

Getzlaff, J. and Oschlies, A.: Pilot Study on Potential Impacts of Fisheries-Induced Changes in Zooplankton Mortality on Marine Biogeochemistry, Global Biogeochemical Cycles, 31, 1656–1673, https://doi.org/10.1002/2017GB005721, 2017.

Hutchings, L.: Fish harvesting in a variable, productive environment — searching for rules

or searching for exceptions?, South African Journal of Marine Science, 12, 297–318, https://doi.org/10.2989/02577619209504708, 1992.

Kiørboe, T.: Zooplankton body composition, Limnology and Oceanography, 58, 1843–1850, https://doi.org/10.4319/lo.2013.58.5.1843, 2013.

Lavery, T. J., Roudnew, B., Gill, P., Seymour, J., Seuront, L., Johnson, G., Mitchell, J. G., And Smetacek, V.: Iron defecation by sperm whales stimulates carbon export in the Southern Ocean, Proceedings of the Royal Society B: Biological Sciences, 277(1699), 3527–3531, https://doi.org/10.1098/rspb.2010.0863, 2010.

McInturf, A. G., Pollack, L., Yang, L. H., and Spiegel, O.: Vectors with autonomy: what distinguishes animal-mediated nutrient transport from abiotic vectors?, Biological Reviews, 94(5), 1761–1773, https://doi.org/10.1111/brv.12525, 2019.

Moriarty, R., and O'Brien, T. D.: Distribution of mesozooplankton biomass in the global ocean, Earth System Science Data, 5(1), 45–55, https://doi.org/10.5194/essd-5-45-2013, 2013.

O'Brien, T. D., and Moriarty, R.: Global distributions of mesozooplankton abundance and biomass - Gridded data product (NetCDF) - Contribution to the MAREDAT World Ocean Atlas of Plankton Functional Types, PANGEA, https://doi.org/10.1594/PANGAEA.785501, 2012.

Ryther, J. H.: Photosynthesis and fish production in the sea, Science, 166, 72–76, 1969.

Varpe, Ø. and Fiksen,Ø.: Meta-ecosystems and biological energy transport from ocean to coast: the ecological importance of herring migration, Oecologia, 146(3), 443–451, https://doi.org/10.1007/s00442-005-0219-9, 2005.

Williams, J. J., Papastamatiou, Y. P., Caselle, J. E., Bradley, D., and Jacoby, D. M. P.: Mobile marine predators: an understudied source of nutrients to coral reefs in an unfished atoll, Proceedings of the Royal Society B: Biological Sciences, 285(1875), 20172456, https://doi.org/10.1098/rspb.2017.2456, 2018.

[Figure]

**Fig. 1.** Same as Fig. 4 of the paper after correcting the averaging weights in experiments A.

[Figure]

**Fig. 2.** Same as Fig. D1 of the paper after correcting the averaging weights in experiments A.

**Fig. 3.** Same as Fig. 4 of the paper after correcting the averaging weights in experiments A and removing the deep water (100 to 1000 m) layer.

[Figure]

**Fig. 4.** Time series from year 21 to year 30 of integrated concentrations (upper 100 m) averaged over space (thick lines), and percentage difference between every point in time and the same date one year later

[Figure]

**Fig. 5.** Comparison of mesozooplankton observations from the global dataset by O'Brien and Moriarty (2012) (see main text for more information), and model large zooplankton averaged over the upper 100 m depth

---

## Author Response (AR1)

Dear editor,

Thank you for your comments. We have now produced a revised version of the manuscript and a document with tracked changes, including the corrections recommended by the referees. We have paid special attention in the reviewed manuscript on expanding on the discussion on the mismatch between model and zooplankton observations. It was mentioned in the response to the second referee that we would exchange the positions of Appendices A and B in the manuscript. However, after carefully revising the manuscript, we decided to leave them as Appendix A: Plankton parameters, and Appendix B: Mesozooplankton evaluation, as this is the order in which they are referenced in the main text. We corrected the value of the initial slope of the P-I curve of small phytoplankton in Table A1. We have also made small language corrections to improve the readability of the manuscript. We have included as supplementary material a figure on the model temporal evolution, and a figure comparing our large zooplankton to an additional mesozooplankton dataset, and a short comment about it in Sect. 2.3. All other changes to the manuscript have been specified in the individual responses to the referees.

Kind regards,

Mariana Hill Cruz